# The Power of Hard Attention Transformers on Data Sequences: A Formal Language Theoretic Perspective

**Pascal Bergsträßer**
RPTU Kaiserslautern-Landau
67663 Kaiserslautern, Germany
bergstraesser@cs.uni-kl.de

**Chris Köcher**
MPI-SWS
67663 Kaiserslautern, Germany
ckoecher@mpi-sws.org

**Anthony Widjaja Lin**
MPI-SWS and RPTU Kaiserslautern-Landau
67663 Kaiserslautern, Germany
awlin@mpi-sws.org

**Georg Zetzsche**
MPI-SWS
67663 Kaiserslautern, Germany
georg@mpi-sws.org

## Abstract

Formal language theory has recently been successfully employed to unravel the power of transformer encoders. This setting is primarily applicable in Natural Language Processing (NLP), as a token embedding function (where a bounded number of tokens is admitted) is first applied before feeding the input to the transformer. On certain kinds of data (e.g. time series), we want our transformers to be able to handle *arbitrary* input sequences of numbers (or tuples thereof) without *a priori* limiting the values of these numbers. In this paper, we initiate the study of the expressive power of transformer encoders on sequences of data (i.e. tuples of numbers). Our results indicate an increase in expressive power of hard attention transformers over data sequences, in stark contrast to the case of strings. In particular, we prove that Unique Hard Attention Transformers (UHAT) over inputs as data sequences no longer lie within the circuit complexity class AC0 (even without positional encodings), unlike the case of string inputs, but are still within the complexity class TC0 (even with positional encodings). Over strings, UHAT without positional encodings capture only regular languages. In contrast, we show that over data sequences UHAT can capture non-regular properties. Finally, we show that UHAT capture languages definable in an extension of linear temporal logic with unary numeric predicates and arithmetics.

## 1 Introduction

Recent years have witnessed the success of transformers [42] in different applications, including natural language processing [12], computer vision [14], speech recognition [13], and time series analysis [45, 46]. In the quest to better understand the ability and limitation of transformers, theoretical investigations have actively been undertaken in the last few years. Among others, *formal language theory* has been successfully applied to reveal deep insights into the expressive power of transformers, e.g., see the recent survey [40] and [2, 3, 8, 15, 20, 21, 30, 32, 38, 39]. In particular, relevant questions pertain to the power of various attention mechanisms, bounded/unbounded precision, positional encoding functions, and interplay between encoders and decoders, among many others.

One common assumption in the formal language theoretic approach to transformers is that the input sequence ranges over a *finite* set $\Sigma$ (called alphabet), which is then to be fed into a transformer after applying a token embedding function of the form $f : \Sigma \to \mathbb{R}^d$. As a by-product, the number of tokens is finite. In certain applications (e.g. time series forecasting [28]), we want our transformers to

38th Conference on Neural Information Processing Systems (NeurIPS 2024).

be able to handle *arbitrary* input sequences of numbers (or tuples thereof) without *a priori* limiting the values of these numbers. Moreover, numbers could be compared using arithmetic and (in)equality, which is not the case for elements of alphabets considered in formal language theory. For this reason, we propose to investigate the expressive power of *transformers over data sequences*, which takes us to the setting of formal language theory over the alphabet $\Sigma = \mathbb{Q}^d$, for some $d \in \mathbb{Z}_{>0}$, e.g., see [4, 11]. That is, *what properties of a sequence of (tuples of) numbers can be recognized by transformers?*

**Connections to circuit complexity.** Existing work has revealed intimate connections between transformers and circuit complexity. In particular, let us consider the following class of transformer encoders that has been the main focus of many recent papers: *Unique Hard Attention Transformers (UHAT)*. Among others, UHAT allows arbitrary positional encoding and an attention mechanism that picks a vector at a unique minimum position in the sequence that maximizes the attention score. It is known that the class of formal languages recognized by UHAT is a strict subset of the circuit complexity class $\mathrm{AC}^0$ (cf. [3, 20, 21]), i.e., each UHAT can be simulated by a family of boolean circuits of constant depth. More concretely, this entails among others that UHAT cannot compute the parity (even/oddness) of the number of occurrences of any given letter $a$ in the input string (for strings over an alphabet containing at least two letters).

**Our first contribution** is that UHAT over data sequences (even without positional encodings) is *no longer* contained in $\mathrm{AC}^0$, unlike the case of finite number of tokens. Instead, we show that UHAT can be captured by the circuit complexity class $\mathrm{TC}^0$, which extends $\mathrm{AC}^0$ circuits with majority gates.

**Theorem 1.** *UHAT with positional encoding over data sequences is in $\mathrm{TC}^0$ but not in $\mathrm{AC}^0$.*

This complexity upper bound allows us to deduce the expressive power of UHAT over data sequences by using complexity theory. For example, since UHAT accepts only $\mathrm{TC}^0$ languages, successfully constructing a UHAT (e.g. through learning) for

$$\mathsf{SQRTSUM} := \left\{ (a_1, b_1), \ldots, (a_n, b_n) \,\middle|\, \sum_{i=1}^{n} \sqrt{a_i} \leq \sum_{i=1}^{n} \sqrt{b_i}, \text{and each } (a_i, b_i) \in \mathbb{Z}_{>0} \times \mathbb{Z}_{>0}, \right\},$$

would constitute a major breakthrough in complexity theory (cf. [1, 18]), i.e., showing that $\mathsf{SQRTSUM}$ is in the complexity class $\mathrm{TC}^0 \subseteq \mathrm{P/poly}$. A byproduct of our proof is that for each length, the set of accepted sequences is a semialgebraic set. This implies, e.g., that the graph $\{(x, e^x) \mid x \in \mathbb{R}\} \subseteq \mathbb{R}^2$ of $x \mapsto e^x$ (viewed as a set of length-1 sequences) is not accepted by UHAT.

**Connection to regular languages over data sequences.** Recent results have revealed surprising connections between regular languages and formal languages recognizable by transformer encoders. In particular, it was proven (cf. [2]) that languages recognizable by UHAT (even *with no positional encodings*) form a strict subset of regular languages, namely those that are "star-free" or, equivalently, definable in First-Order Logic (FO), or Linear Temporal Logic (LTL). With positional encodings, similar connections hold, by extending the logics with *unary numerical predicates* (cf. [2, 3]).

To investigate whether such connections extend to data sequences, we bring forth *formal languages theory over infinite alphabets* (cf., [4, 11]), which has been an active research field in the last decade or so with applications to programming languages and databases (to name a few), e.g., see [11, 26, 43]. **Our second contribution** is a language over data sequences recognizable by UHAT without positional encodings that lies beyond existing formal models over infinite alphabets (in particular, "regular" ones). This shows the strength of UHAT over data sequences even without positional encodings, in stark contrast to the case of finite alphabets.

**Theorem 2.** *There is a non-regular language over $\Sigma = \mathbb{Q}^d$ that is accepted by masked UHAT with no positional encoding.*

Finally, to better understand languages over data sequences recognizable by UHAT, **our third contribution** is to show how UHAT can recognize languages definable by the so-called *Locally Testable LTL* $((\mathrm{LT})^2\mathrm{L})$, which extends LTL with unary numerical predicates and *local arithmetic tests* for fixed-size windows over the input sequence. Using $(\mathrm{LT})^2\mathrm{L}$, we can see *at a glance* what can be expressed by UHAT over data sequences. For one, this includes the well-known *Simple Moving Averages*. As another example, using $(\mathrm{LT})^2\mathrm{L}$ it can be easily shown that UHAT can capture linear recurrence sequences considered in the famous Skolem problem and discrete linear dynamical systems [25, 27, 29], i.e., sequences of the form $\boldsymbol{x}, A\boldsymbol{x}, \ldots, A^n\boldsymbol{x}$ such that $n \geq 0$ is minimal with $\boldsymbol{y}A^n\boldsymbol{x} = 0$ where $\boldsymbol{y} \in \mathbb{Q}^{1 \times d}$ and $A \in \mathbb{Q}^{d \times d}$ are fixed and $\boldsymbol{x} \in \mathbb{Q}^d$.

**Theorem 3.** *Every $(LT)^2L$-definable language is accepted by UHAT with positional encoding.*

**Technical challenges.** Obtaining our results poses several challenges. First, for the $TC^0$ upper bound, we need to use Boolean (constant depth) circuits to simulate UHATs, in which real constants can occur (in affine transformations or positional encodings). While in $TC^0$, it is known that majority gates can be used to perform multiplication of rationals [7], arithmetic with reals requires infinite precision and cannot be done with Boolean circuits. To this end, we compute rational approximations of reals accurate enough to preserve the acceptance condition for inputs up to a particular length $n$.

Here, a naive attempt would be to replace each real occurring in the UHAT in affine transformations and the positional encoding by some rational approximation. However, this is not possible, meaning *any* rational approximation would change the behavior on input sequences of length $n = 3$, even for low-dimensional vectors with entries in $\{0, 1\}$. Indeed, there is a UHAT involving real numbers $\alpha, \beta$ that accepts a simple sequence of $\{0, 1\}$-vectors if and only if $\alpha\beta = 2$ and $\alpha = \beta$, i.e. $\alpha = \beta = \sqrt{2}$. Thus, $\alpha$ and $\beta$ cannot be replaced by rationals, even for very short inputs (see Appendix A for details).

Instead, we show that a UHAT can be translated into a small Boolean combination of polynomial inequalities. This format has the advantage that—as we show using convex geometry—the real coefficients of those polynomials *can* be replaced by suitably chosen rational numbers. In turn, the layer-by-layer construction of these polynomial inequalities requires a carefully chosen data structure to encode the function computed by a sequence of transformer layers. For example, we show that the resulting Boolean combinations of polynomial inequalities have a bound on the number of alternations between conjunctions and disjunctions, which is crucial for constructing $TC^0$-circuits.

Another key challenge occurs in the translation from $(LT)^2L$ to UHATs: In the inductive construction, we need to represent truth values using reals in $[0, 1]$. To implement negation, we use a UHAT gadget (with positional encodings) that can normalize these truth values to $\{0, 1\}$.

**Notation.** In the sequel, we assume some background from computational complexity, in particular circuit complexity (see the book [44]). In particular, we use the circuit complexity class $AC^0$, which defines a class of problems that are computable by a nonuniform family of constant-depth boolean circuits, where each gate permits an unbounded fan-in (i.e. arbitrary many inputs). Similarly, the complexity class $TC^0$ is an extension of $AC^0$, where majority gates are allowed. It is well-known that $AC^0 \subsetneq TC^0$. Assuming *uniformity*, both $AC^0$ and $TC^0$ are contained in the class of problems solvable in polynomial-time. For *nonuniformity*, these classes are contained in the complexity class P/poly, which admits (nonuniform) polynomial-size circuits. It is a long-standing open problem whether numerical analysis (e.g. square-root-sum) is in P/poly, e.g., see [1].

## 2 Transformer encoders and their languages

In the following, we adapt the setting of formal language theory (see [3, 21, 40]) to data sequences. For a vector $\boldsymbol{a} = (a_1, \dots, a_d)$ we write $\boldsymbol{a}[i, j] := (a_i, \dots, a_j)$ for all $1 \leq i \leq j \leq d$ and if $i = j$, we simply write $\boldsymbol{a}[i]$. For a set $S$ we denote the set of (potentially empty) sequences of elements from $S$ by $S^*$. We write $S^+$ for the restriction to non-empty sequences. We consider languages $L$ over the infinite alphabet $\Sigma = \mathbb{Q}^d$, for some integer $d > 0$. That is, $L$ is a set of sequences of $d$-tuples over rational numbers. We will define a UHAT (similarly as in previous papers that study formal language theoretic perspectives) as a length preserving map $(\mathbb{Q}^d)^* \to (\mathbb{R}^e)^*$.

**Standard encoder layer with unique hard attention.** A standard encoder layer is defined by three affine transformations $A, B \colon \mathbb{R}^d \to \mathbb{R}^d$ and $C \colon \mathbb{R}^{2d} \to \mathbb{R}^e$. For a sequence $\boldsymbol{v}_1, \dots, \boldsymbol{v}_n \in \mathbb{R}^d$ with $n \geq 1$ we define the *attention vector* at position $i \in [1, n]$ as $\boldsymbol{a}_i := \boldsymbol{v}_j$ with $j \in [1, n]$ minimal such that the *attention score* $\langle A\boldsymbol{v}_i, B\boldsymbol{v}_j \rangle$ is maximized. The layer outputs the sequence $C(\boldsymbol{v}_1, \boldsymbol{a}_1), \dots, C(\boldsymbol{v}_n, \boldsymbol{a}_n)$.

**ReLU encoder layer.** A ReLU layer for some $k \in [1, d]$ on input $\boldsymbol{v}_1, \dots, \boldsymbol{v}_n \in \mathbb{R}^d$ applies the ReLU function to the $k$-th coordinate of each $\boldsymbol{v}_i$, i.e. it outputs the sequence $\boldsymbol{v}'_1, \dots, \boldsymbol{v}'_n$ where $\boldsymbol{v}'_i := (\boldsymbol{v}_i[1, k-1], \max\{0, \boldsymbol{v}_i[k]\}, \boldsymbol{v}_i[k+1, n])$. [Equivalently, one could instead allow a feed-forward network at the end of an encoder layer (see [3, 21, 40]).]

**Transformer encoder.** A *unique hard attention transformer encoder* (UHAT) is a repeated application of standard encoder layers with unique hard attention and ReLU encoder layers. Clearly, using an

alternation of standard layers and ReLU layers, we can assume that the output of a UHAT layer is an arbitrary composition of affine transformations and component-wise ReLU application. In particular, these compositions may use the functions max and min.

**Languages accepted by UHATs.** The notion of "languages" accepted by a UHAT (i.e. a set of accepted sequences) can be defined, depending on whether a *positional encoding* is permitted. If it is permitted, a language $L \subseteq (\mathbb{Q}^d)^+$ is accepted by a UHAT $T$ if and only if there exists a positional encoding function $p \colon \mathbb{N} \times \mathbb{N} \to \mathbb{R}^s$ and an acceptance vector $\boldsymbol{t} \in \mathbb{R}^e$ such that on every sequence

$$(p(1, n+1), \boldsymbol{v}_1), \dots, (p(n, n+1), \boldsymbol{v}_n), (p(n+1, n+1), \boldsymbol{0}) \tag{1}$$

$T$ outputs a sequence $\boldsymbol{v}_1', \dots, \boldsymbol{v}_{n+1}' \in \mathbb{R}^e$ with $\langle \boldsymbol{t}, \boldsymbol{v}_1' \rangle > 0$ if and only if $(\boldsymbol{v}_1, \dots, \boldsymbol{v}_n) \in L$. Note that if $T_1$ and $T_2$ are UHATs with positional encoding that realize functions $f_1 \colon (\mathbb{Q}^d)^* \to (\mathbb{Q}^e)^*$ and $f_2 \colon (\mathbb{Q}^e)^* \to (\mathbb{Q}^r)^*$, then there is a UHAT $T_2 \circ T_1$ with positional encoding that realizes the composition $f_2 \circ f_1$ by using a positional encoding that combines the positional encodings of $T_1, T_2$.

In the above definition we appended an additional zero vector to the end of the input. Over finite alphabets it is often assumed that the input sequence is extended with a special unique end-of-input marker (e.g. see [21, 40]). When the input is a sequence of (tuples of) numbers, if we allow positional encoding, then the zero vector at the end of the input can be turned into a unique vector marking the end of the input (see Section 5). Without positional encoding, however, we have to explicitly make the zero vector at the end of the input unique. That is, a UHAT without positional encoding is initialized with the sequence $(1, \boldsymbol{v}_1), \dots, (1, \boldsymbol{v}_n), \boldsymbol{0} \in \mathbb{Q}^{d+1}$ instead; this ensures, among others, that the end-of-input marker does not appear in the actual input.

In the definition of a standard encoder layer the attention vector at position $i \in [1, n]$ can be any vector in the sequence $\boldsymbol{v}_1, \dots, \boldsymbol{v}_n$. Using *masking*, one can restrict the attention vector to vectors of certain positions. A UHAT with *past masking* restricts the attention vector $\boldsymbol{a}_i$ at position $1 \leq i < n$ to be contained in the subsequence $\boldsymbol{v}_{i+1}, \dots, \boldsymbol{v}_n$ and at position $n$ to $\boldsymbol{a}_n := \boldsymbol{v}_n$.

# 3 UHAT and TC$^0$

In this section, we prove Theorem 1. First, we show that all languages of UHAT (even with positional encoding) belong to the class TC$^0$. Then, we show that there is a UHAT (even without positional encoding) whose language is TC$^0$-hard under AC$^0$-reductions. We begin with the proof that all UHAT languages belong to TC$^0$.

**Input encoding** A language $L \subseteq \Sigma^*$ over a finite alphabet $\Sigma$ belongs to TC$^0$ if for every input length $n$, there is a circuit of size polynomial $n$, such that the circuit consists of input gates (for each input position $i$, and each letter $a \in \Sigma$, there is a gate that evaluates to "true" if and only if position $i$ of the input words carries an $a$), Boolean gates (computing the AND, OR, or NOT function) and majority gates (evaluating to true if more than half of their input wires carry true). Here, AND, OR, and majority gates can have arbitrary fan-in. In order to define what it means that a language $L \subseteq (\mathbb{Q}^d)^+$ belongs to TC$^0$, we need to specify an encoding as finite-alphabet words. To this end, we encode a sequence $\boldsymbol{u}_1, \dots, \boldsymbol{u}_n$ with $\boldsymbol{u}_i \in \mathbb{Q}^d$ as a string $v_1 \# \cdots \# v_n$, where $v_i = p_1/q_1 \square \cdots \square p_d/q_d$ with $p_j, q_j \in \{-, 0, 1\}^*$. Here, $v_i \in \{-, /, \square, 0, 1\}^*$ represents the vector $\boldsymbol{u}_i \in \mathbb{Q}^d$ such that $\boldsymbol{u}_i[j] = \frac{a_j}{b_j}$, $a_j, b_j \in \mathbb{Z}$, and $p_j, q_j \in \{-, 0, 1\}^*$ are the binary expansions of $a_j$ and $b_j$.

**Remark 4.** *The main challenge in proving Theorem 1 is that the constants appearing in a UHAT can be real numbers. These can in general not be avoided: There are UHAT languages over $\Sigma = \mathbb{Q}$ (even without positional encoding) that cannot be accepted by UHAT with rational constants (even with positional encoding). For example, for every real number $r > 0$, one can[1] construct a UHAT for the language $L_r$ of all sequences over $\Sigma = \mathbb{Q}$ where the first letter is $> r$. Note that $L_r \neq L_s$ for any $r, s > 0$, $r \neq s$. However, there are clearly only countably many languages over $\Sigma = \mathbb{Q}$ accepted by UHAT with rational constants where membership only depends on the first letter[2]. Thus, there are uncountably many real $r > 0$ such that $L_r$ is not accepted by a UHAT with rational constants.*

---

[1] First, use a standard encoding layer to transform the sequence $(x_1, \dots, x_n) \in \Sigma^n$ into $(r^{-1}x_1 - 1, \dots, r^{-1}x_n - 1) \in \Sigma^n$. Then, accept if and only if the left-most vector is $> 0$.

[2] This is because all employed affine transformations (which are rational matrices), but also the values $p(1, 2), p(2, 2) \in \mathbb{Q}$ of the positional encoding can be described using finitely many bits.

The construction of TC$^0$ circuits comprises three steps. In Step I, we show that the set of accepted length-$n$ sequences can be represented by a Boolean combination of polynomial inequalities. Importantly, (i) this representation, called "polynomial constraints" is polynomial-sized in $n$, and (ii) the number of alternations between conjunction and disjunction is bounded (i.e. independent of $n$). The polynomials in this representations can still contain real coefficients. In Step II, we show that if we restrict the input not only to length-$n$ sequences, but to rational numbers of size $\leq m$, then we can replace all real coefficients of our polynomials by rationals of size polynomial in $m$ and $n$, without changing the language (among vectors of size $\leq m$). In Step III, we implement a TC$^0$ circuit. Here, it is important that the number of alternations between conjunctions and disjunctions in our polynomial constraints is bounded, because the depth of the circuit is proportional to this number of alternations.

**Step I: UHAT as polynomials**    We first consider a formalism to describe a set of sequences over $\mathbb{Q}^d$. We consider such sequences $(\boldsymbol{x}_1, \ldots, \boldsymbol{x}_n)$ of length $n$, where $\boldsymbol{x}_i \in \mathbb{Q}^d$ for each $i$. In this case, we also abbreviate $\bar{\boldsymbol{x}} = (\boldsymbol{x}_1, \ldots, \boldsymbol{x}_n)$. A *polynomial constraint* (PC) is a positive Boolean combination (i.e., without negation) of constraints of the form $p(\bar{\boldsymbol{x}}) > 0$ or $p(\bar{\boldsymbol{x}}) \geq 0$, where $p \in \mathbb{R}[X_1, \ldots, X_{d \cdot n}]$ is a polynomial with real coefficients. Here, plugging $\bar{\boldsymbol{x}} \in (\mathbb{Q}^d)^n$ into $p$ is defined by assigning the $d \cdot n$ rational numbers in $\bar{\boldsymbol{x}}$ to the $d \cdot n$ variables $X_1, \ldots, X_{d \cdot n}$. The PC $\alpha$ *accepts* a sequence of vectors $\bar{\boldsymbol{x}} \in (\mathbb{Q}^d)^n$, if the Boolean formula evaluates to true when plugging $\bar{\boldsymbol{x}}$ into the polynomials $p$ in $\alpha$. The set of accepted sequences is denoted by $[\![\alpha]\!]$. Now let $a \in \mathbb{N}$. A PC has $a$ *alternations* if the positive Boolean combination has $a$ alternations between disjunctions and conjunctions.

In the following, a *constrained polynomial representation (CPR)* can be used to compute from a sequence of inputs $(\boldsymbol{x}_1, \ldots, \boldsymbol{x}_n)$ with $\boldsymbol{x}_1, \ldots, \boldsymbol{x}_n \in \mathbb{R}^{d'}$ a new sequence of outputs $(\boldsymbol{y}_1, \ldots, \boldsymbol{y}_n)$ with $\boldsymbol{y}_1, \ldots, \boldsymbol{y}_n \in \mathbb{R}^d$. Formally, a CPR comprises for each $i \in \{1, \ldots, n\}$ a sequence $(\varphi_1, D_1), \ldots, (\varphi_{s_i}, D_{s_i})$ of pairs $(\varphi_j, D_j)$, where each pair $(\varphi_j, D_j)$ is a "conditional assignment": each $(\varphi_j, D_j)$ tells us that if the condition $\varphi_j$ is satisfied, then we return $D_j(\bar{\boldsymbol{x}})$. More precisely: (i) each $\varphi_j$ is a polynomial constraint where all polynomials have degree $\leq 2$, (ii) for any $j \neq m$, the constraints $\varphi_j$ and $\varphi_m$ are mutually exclusive, and (iii) each $D_j \colon \mathbb{R}^{d' \cdot n} \to \mathbb{R}^d$ is an affine transformation. Because of their role as conditional assignments, we also write $\varphi_j \to D_j$ for such pairs. For $a \in \mathbb{N}$, we say that the CPR is $a$-*alternation-bounded* if each of the formulas $\varphi_j$ has at most $a$ alternations. A CPR as above computes a function $\mathbb{R}^{d' \cdot n} \to \mathbb{R}^{d \cdot n}$: Given $\bar{\boldsymbol{x}} = (\boldsymbol{x}_1, \ldots, \boldsymbol{x}_n)$ with $\boldsymbol{x}_1, \ldots, \boldsymbol{x}_n \in \mathbb{R}^{d'}$, it computes the sequence $(\boldsymbol{y}_1, \ldots, \boldsymbol{y}_n)$ if for every $i \in \{1, \ldots, n\}$, we have $\boldsymbol{y}_i = D_j(\bar{\boldsymbol{x}})$, provided that $j$ is the (in case of existence uniquely determined) index for which $\varphi_j(\bar{\boldsymbol{x}})$ is satisfied. The *size* of PCs and CPRs are their bit lengths (see Appendix B for details).

**Proposition 5.** *Fix a UHAT with positional encoding and $\ell$ layers. For any given sequence length $n$, there exists a polynomial-sized PC $\alpha$ with $O(\ell)$ alternations such that $[\![\alpha]\!]$ equals the set of accepted sequences of length $n$.*

Note that Proposition 5 implies that the set of sequences of each length $n$ is a semialgebraic set [31]. The proof is by induction on the number of layers, which requires a slight strengthening:

**Lemma 6.** *Fix a UHAT with positional encoding and $\ell$ layers. For any given sequence length $n$, one can construct in polynomial time an $O(\ell)$-alternation-bounded CPR computing the function $\mathbb{R}^{d \cdot (n+1)} \to \mathbb{R}^{e \cdot (n+1)}$ computed by the UHAT.*

*Proof.* We prove the statement by induction on the number of layers. First, we consider the positional encoding $p \colon \mathbb{N} \times \mathbb{N} \to \mathbb{R}^d$ as some affine transformations $P_i \colon \mathbb{R}^{d \cdot (n+1)} \to \mathbb{R}^d$ mapping the input sequence $\bar{\boldsymbol{x}}$ to $\boldsymbol{x}_i + p(i, n+1)$. Then we obtain a CPR with $\top \to P_i$ for each $1 \leq i \leq n+1$. Now, suppose the statement is shown for $\ell$ layers and consider a UHAT with $\ell + 1$ layers. Suppose that the first $\ell$ layers of our UHAT compute a function $\mathbb{R}^{d' \cdot (n+1)} \to \mathbb{R}^{d \cdot (n+1)}$, and the last layer computes a function $\mathbb{R}^{d \cdot (n+1)} \to \mathbb{R}^{e \cdot (n+1)}$. By induction, we have a polynomial size CPR consisting of conditional assignments $\varphi_{i,k} \to D_{i,k}$ for every $i \in \{1, \ldots, n+1\}$ and $1 \leq k \leq s_i$. Here, each $D_{i,j}$ is an affine transformation $\mathbb{R}^{d' \cdot (n+1)} \to \mathbb{R}^d$.

Let us first consider the case that the last layer of our UHAT is a standard encoder layer. For each $(i, I, j, J) \in \{1, \ldots, n+1\}^4$, we build the conditional assignment using the formula $\psi_{i,I,j,J}$:

$$\bigwedge_{m=1}^{j-1} \left( \bigvee_{M=1}^{s_m} \varphi_{m,M} \wedge p_{i,I,j,J,m,M}(\bar{\boldsymbol{x}}) > 0 \right) \wedge \bigwedge_{m=j+1}^{n+1} \left( \bigvee_{M=1}^{s_m} \varphi_{m,M} \wedge p_{i,I,j,J,m,M}(\bar{\boldsymbol{x}}) \geq 0 \right)$$

where $p_{i,I,j,J,m,M}(\bar{\boldsymbol{x}})$ is the polynomial $\langle AD_{i,I}\bar{\boldsymbol{x}},\ BD_{j,J}\bar{\boldsymbol{x}} - BD_{m,M}\bar{\boldsymbol{x}}\rangle$. Then, the conditional assignment is $\varphi_{i,I} \wedge \varphi_{j,J} \wedge \psi_{i,I,j,J} \to C(D_{i,I}\bar{\boldsymbol{x}}, D_{j,J}\bar{\boldsymbol{x}})$. Here, the idea is that (i) $\varphi_{i,I}$ expresses that the $I$-th conditional assignment was used to produce the $i$-th cell in the previous layer, (ii) $\varphi_{j,J}$ says the $J$-th conditional assignment was used to produce the $j$-th vector in the previous layer, and (iii) $\psi_{i,I,j,J}$ says the vector $\boldsymbol{x}_j$ yields the maximal attention score for the input $\boldsymbol{x}_i$, meaning (iii-a) for all positions $m < j$, $\boldsymbol{x}_m$ has a lower score than $\boldsymbol{x}_j$ (left parenthesis), and (iii-b) for all positions $m > j$, $\boldsymbol{x}_m$ has at most the score of $\boldsymbol{x}_j$ (right parenthesis). In (iii-a) and (iii-b), we first find the index $M$ of the conditional assignment used to produce the $m$-th cell in the previous layer. Note that then indeed, all the PCs $\psi_{i,I,j,I}$ are mutually exclusive. Moreover, the polynomials $\langle AD_{i,I}\bar{\boldsymbol{x}},\ BD_{j,J}\bar{\boldsymbol{x}} - BD_{m,M}\bar{\boldsymbol{x}}\rangle$ have indeed degree 2 and are of size polynomial in $n$. Furthermore, if the assignments $\varphi_{i,k} \to D_{i,k}$ had at most $a$ alternations, then the new assignments have at most $a + 3$ alternations. Finally, the case of ReLU layers is straightforward (see Appendix B). $\qquad\square$

Finally, the proof of Proposition 5 is straightforward: from the constructed CPR in Lemma 6 we obtain the polynomial constraint $\bigvee_{J=1}^{s_1} \varphi_{1,J} \wedge \langle \boldsymbol{t}, D_{1,J}\bar{\boldsymbol{x}}\rangle > 0$ with a bounded number of alternations.

**Step II: Replace real coefficients by rationals**  In our proof, the key step is to replace the real coefficients in the PC by rational coefficients so that the rational PC will define the same set of rational sequences, up to some given size. Let us make this precise. We denote by $\mathbb{Q}_{\leq m} = \{a \in \mathbb{Q} \mid \|a\|_2 \leq m\}$ the set of all rational numbers of size at most $m$. A polynomial constraint is *rational* if all the polynomials occurring in it have rational coefficients.

**Proposition 7.** *For every $m \in \mathbb{N}$ and every PC $\alpha$ with polynomials having $n$ variables, there exists a rational PC $\alpha'$ of polynomial size such that $[\![\alpha]\!] \cap \mathbb{Q}_{\leq m}^n = [\![\alpha']\!] \cap \mathbb{Q}_{\leq m}^n$.*

Proving Proposition 7 requires the following technical lemma, for which we introduce some notation. For two vectors $\boldsymbol{u}, \boldsymbol{v} \in \mathbb{R}^t$ and $m \in \mathbb{N}$, we write $\boldsymbol{u} \sim_m \boldsymbol{v}$ if for every $\boldsymbol{w} \in \mathbb{Q}^t$ and every $z \in \mathbb{Q}$ with $\|\boldsymbol{w}\|_2, \|z\|_2 \leq m$, we have (i) $\langle \boldsymbol{w}, \boldsymbol{u}\rangle \geq z$ if and only if $\langle \boldsymbol{w}, \boldsymbol{v}\rangle \geq z$ and (ii) $\langle \boldsymbol{w}, \boldsymbol{u}\rangle > z$ if and only if $\langle \boldsymbol{w}, \boldsymbol{v}\rangle > z$. In other words, we have $\boldsymbol{u} \sim_m \boldsymbol{v}$ if and only if $\boldsymbol{u}$ and $\boldsymbol{v}$ satisfy exactly the same inequalities with rational coefficients of size at most $m$.

**Lemma 8.** *For every $\boldsymbol{c} \in \mathbb{R}^t$ and $m \in \mathbb{N}$, there is a $\boldsymbol{c}' \in \mathbb{Q}^t$ with $\|\boldsymbol{c}'\|_2 \leq (mt)^{O(1)}$ and $\boldsymbol{c} \sim_m \boldsymbol{c}'$.*

**Remark 9.** *For proving Lemma 8, it is not sufficient to pick a rational $\boldsymbol{c}'$ with $\|\boldsymbol{c}' - \boldsymbol{c}\| < \varepsilon$ for some small enough $\varepsilon$. For example, note that if in some coordinate, $\boldsymbol{c}$ contains a rational number of size $\leq m$, then in this coordinate, $\boldsymbol{c}'$ and $\boldsymbol{c}$ must agree exactly for $\boldsymbol{c} \sim_m \boldsymbol{c}'$ to hold.*

Before we prove Lemma 8, let us see how to deduce Proposition 7: in a PC $\alpha$ we understand each polynomial $p(X_1, \ldots, X_n)$ as a scalar product $\langle \boldsymbol{w}, \boldsymbol{u}\rangle$ where $\boldsymbol{w}$ constains only variables and $\boldsymbol{u}$ consists of all coefficients. Then Lemma 8 yields a vector $\boldsymbol{v} \sim_{2m} \boldsymbol{u}$ containing only rationals with the same behavior as $\boldsymbol{u}$. From this we finally obtain polynomials having only rational coefficients, which also proves Proposition 7. A detailed proof of Proposition 7 can be found in Appendix B.

In the proof of Lemma 8, we use the following fact about solution sizes to systems of inequalities.

**Lemma 10.** *Let $A \in \mathbb{Q}^{k \times n}$, $A' \in \mathbb{Q}^{\ell \times n}$, $\boldsymbol{z} \in \mathbb{Q}^k$, $\boldsymbol{z}' \in \mathbb{Q}^\ell$ with $\|A\|_2, \|A'\|_2, \|\boldsymbol{z}\|_2, \|\boldsymbol{z}'\|_2 \leq m$. If the inequalities $A\boldsymbol{x} \gg \boldsymbol{z}$ and $A'\boldsymbol{x} \geq \boldsymbol{z}'$ have a solution in $\mathbb{R}^n$, then they have a solution $\boldsymbol{r} \in \mathbb{Q}^n$ with $\|\boldsymbol{r}\|_2 \leq (mn)^{O(1)}$.*

We prove Lemma 10 in the appendix. The proof idea is the following. By standard results about polyhedra, the set of vectors $\boldsymbol{x}$ satisfying $A\boldsymbol{x} \geq \boldsymbol{z}$ and $A'\boldsymbol{x} \geq \boldsymbol{z}'$ can be written as the convex hull of some finite set $X = \{\boldsymbol{x}_1, \ldots, \boldsymbol{x}_s\}$, plus the cone generated by some finite set $\{\boldsymbol{y}_1, \ldots, \boldsymbol{y}_t\}$. Here, the vectors in $X \cup Y$ are all rational and of polynomial size. By the Carathéodory Theorem, the real solution $\boldsymbol{s} \in \mathbb{R}^n$ to $A\boldsymbol{s} \gg \boldsymbol{z}$ and $A'\boldsymbol{s} \geq \boldsymbol{z}'$ belongs to the convex hull of $n$ elements of $X$, plus a conic combination of $n$ elements of $Y$. We then argue that by taking the barycenter of those $n$ elements of $X$, plus the sum of the $n$ elements of $Y$ gives a rational vector $\boldsymbol{r} \in \mathbb{Q}^n$ with $A\boldsymbol{r} \gg \boldsymbol{z}$ and $A'\boldsymbol{r} \geq \boldsymbol{z}'$. The full proof of Lemma 10 is in Appendix B.4. To prove Lemma 8, given $\boldsymbol{c} \in \mathbb{R}^n$, we set up a system of (exponentially many) inequalities of polynomial size so that the solutions are exactly the vectors $\boldsymbol{d}$ with $\boldsymbol{d} \sim_m \boldsymbol{c}$. The solution provided by Lemma 10 is the desired $\boldsymbol{c}'$ (see Appendix B.5).

**Step III: Constructing $TC^0$ circuits**  It is now straightforward to translate a polynomial-sized CPR with rational coefficients and bounded alternations into a $TC^0$ circuit:

**Proposition 11.** *Every language accepted by a UHAT with positional encoding is recognized by a family of circuits in $TC^0$.*

We now show that the $TC^0$ upper bound is tight: There is a UHAT whose language is $TC^0$-hard under $AC^0$ reductions. In particular, this language is not in $AC^0$, since $AC^0$ is strictly included in $TC^0$.

**Proposition 12.** *There is a $TC^0$-complete language that is accepted by a UHAT, even without positional encoding and masking, but is not recognized by any family of circuits in $AC^0$.*

*Proof.* As shown by Buss [6, Corollary 3], the problem of deciding whether $ab = c$ for given binary encoded integers $a, b, c \in \mathbb{Z}$ is $TC^0$-complete under $AC^0$-reductions. Since $ab = c$ if and only if $ab > c - 1$ and $-ab > -(c + 1)$, the problem of deciding $ab > c$ is also $TC^0$-complete. We exibit a UHAT such that the problem of deciding $ab > c$ can be $AC^0$-reduced to membership in the language.

It suffices to define a UHAT $T$ that accepts a language $L \subseteq (\mathbb{Q}^2)^+$ such that for all $r, s \in \mathbb{Q}$ we have that $(r, s) \in L$ if and only if $r > s$. Then we can reduce the test $ab > c$ to checking whether $(a, \frac{c}{b}) \in L$. Note that formally, $\frac{c}{b}$ is represented as a string containing the binary encodings of $c$ and $b$ separated by a special symbol. The UHAT $T$ is by definition initialized with the sequence $(1, r, s), (0, 0, 0) \in \mathbb{Q}^3$ since we only have to consider the accepted language restricted to sequences of length 1. It can directly check that $r - s > 0$ using the acceptance vector $\boldsymbol{t} := (0, 1, -1)$. $\qquad\square$

## 4 UHAT and regular languages over infinite alphabets

It was shown by Angluin et al. [2] that UHATs with no positional encoding on binary input strings accept only regular languages, even if masking is allowed. We show that UHATs with masking over data sequences can recognize "non-regular" languages over infinite alphabet (Theorem 2). More precisely, a standard notion of regularity over the alphabet $\Sigma = \mathbb{Q}^d$ is that of *symbolic automata* (see the CACM article [11]), since it extends and shares all nice closure and algorithmic properties of finite automata over finite alphabets, while at the same time permitting arithmetics. Intuitively, a transition rule in a symbolic automaton is of the form $p \to_\varphi q$, where $\varphi$ represents the (potentially infinite) set $S \subseteq \mathbb{Q}^d$ of solutions to an arithmetic constraint $\varphi$ (e.g. $2x = y$ represents $\{(n, 2n) : n \in \mathbb{Q}\}$). The meaning of such a transition rule is: move from state $p$ to state $q$ by reading any $a \in S$.

To prove Theorem 2, we define the language

$$\mathsf{Double} := \{(r_1, \ldots, r_n) \in \mathbb{Q}^n \mid n \geq 1 \text{ and } 2r_i < r_{i+1} \text{ for all } 1 \leq i < n\}$$

of sequences of rational numbers where each number is more than double the preceding number.

**Lemma 13.** *UHAT with past masking and without positional encoding can recognize* $\mathsf{Double}$.

*Proof.* Given an input sequence $\boldsymbol{u}_1, \ldots, \boldsymbol{u}_{n+1} = (1, r_1), \ldots, (1, r_n), (0, 0) \in \mathbb{Q}^2$, we need to check that for all pairs $1 \leq i < j < n + 1$, we have $2 \cdot r_i < r_j$. To this end, a first standard encoder layer uses the differences $2r_i - r_j$ as attention scores—except for $j = n + 1$, where the attention score will be 0. This is achieved by setting the attention score for positions $i, j$ to $2\boldsymbol{u}_i[2] \cdot \boldsymbol{u}_j[1] - \boldsymbol{u}_j[2]$. Indeed, this evaluates to $2r_i - r_j$ for $i < j < n + 1$, and to 0 for $i < j = n + 1$. In particular, for a position $i \in [1, n]$, the attention score is maximized at $j = n + 1$ if and only if $2r_i < r_j$ for all $j \in [i + 1, n]$.

The output vector $\boldsymbol{v}_i$ at position $i$ is then set to $\boldsymbol{a}_i[1]$, where $\boldsymbol{a}_i$ is the attention vector at position $i$. Thus, the output vector has dimension 1, and for each $i \in [1, n + 1]$, we have $\boldsymbol{v}_i = 0$ if and only if $2 \cdot r_i < r_j$ holds for all $j \in [i + 1, n]$.

In a second standard encoder layer we now check whether all $\boldsymbol{v}_i$ have value 0. To this end, we choose for $i < j \leq n + 1$ the attention score $\boldsymbol{v}_j$. Let $\boldsymbol{b}_i$ denote the attention vector at position $i$. Then $\boldsymbol{b}_i = 0$ iff $\boldsymbol{v}_{i+1}, \ldots, \boldsymbol{v}_n$ are all 0. We then output $\boldsymbol{w}_i = 1 - (\boldsymbol{v}_i + \boldsymbol{b}_i)$, which is positive if and only if $\boldsymbol{v}_i = \cdots = \boldsymbol{v}_{n+1} = 0$. Finally, with the acceptance vector $\boldsymbol{t} = 1$ we accept if and only if $\boldsymbol{w}_1 > 0$, which is equivalent to $\boldsymbol{v}_1 = \cdots = \boldsymbol{v}_{n+1} = 0$. As we saw above, the latter holds if and only if $2r_i < r_j$ for all $i, j$ with $1 \leq i < j < n + 1$. $\qquad\square$

The proof of non-regularity of Double is easy (see Appendix C). One could also easily show that Double cannot be recognized by other existing models in the literature of formal language theory over infinite alphabets, e.g., register automata [4, 9, 24, 36, 41], variable/parametric automata [16, 19, 23],

and data automata variants [5, 17]. For example, for register automata over $(\mathbb{Q}; <)$ (see the book [4]), one could use the result therein that data sequences accepted by such an automaton are closed under any order-preserving map of the elements in the sequence (e.g., if $1, 2, 3$ is accepted, then so is $10, 11, 20$), which is not satisfied by Double.

## 5   Logical languages accepted by UHAT

In this section we show that an extension of linear temporal logic (LTL) with linear rational arithmetic (LRA) and unary numerical predicates is expressible in UHAT over data sequences (Theorem 3). A formula of dimension $d > 0$ in *locally testable LTL* ($(LT)^2L$) has the following syntax:

$$\varphi ::= \psi_k(\boldsymbol{x}_1, \ldots, \boldsymbol{x}_{k+1}) \mid \Theta \mid \neg\varphi \mid \varphi \vee \varphi \mid X\varphi \mid \varphi U\varphi$$

Here, $\psi_k$ for $k \geq 0$ is an atom in LRA over the $d$-dimensional vectors of variables $\boldsymbol{x}_i$ of the form $\langle \boldsymbol{a}, (\boldsymbol{x}_1, \ldots, \boldsymbol{x}_{k+1}) \rangle + b > 0$ where $\boldsymbol{a} \in \mathbb{Q}^{d(k+1)}$ and $b \in \mathbb{Q}$. Intuitively, $\psi_k$ allows one to check the values in the sequence with $k$ "lookaheads". Furthermore, $\Theta$ is a *unary numerical predicate*, i.e. a family of functions $\theta_n \colon \{1, \ldots, n\} \to \{0, 1\}$ for all $n \geq 1$. We define the satisfaction of an $(LT)^2L$ formula $\varphi$ over a sequence $\bar{\boldsymbol{v}} = (\boldsymbol{v}_1, \ldots, \boldsymbol{v}_n)$ of vectors in $\mathbb{Q}^d$ at position $i \in [1, n]$, written $(\bar{\boldsymbol{v}}, i) \models \varphi$, inductively as follows (omitting negation and disjunction):

- $(\bar{\boldsymbol{v}}, i) \models \psi_k(\boldsymbol{x}_1, \ldots, \boldsymbol{x}_{k+1})$ iff $i \leq n - k$ and $\psi_k(\boldsymbol{v}_i, \ldots, \boldsymbol{v}_{i+k})$
- $(\bar{\boldsymbol{v}}, i) \models \Theta$ iff $\theta_n(i) = 1$
- $(\bar{\boldsymbol{v}}, i) \models X\varphi$ iff $i < n$ and $(\bar{\boldsymbol{v}}, i + 1) \models \varphi$
- $(\bar{\boldsymbol{v}}, i) \models \varphi U\psi$ iff there is $j \in [i, n]$ with $(\bar{\boldsymbol{v}}, j) \models \psi$ and $(\bar{\boldsymbol{v}}, k) \models \varphi$ for all $k \in [i, j - 1]$

We write $L(\varphi) := \{\bar{\boldsymbol{v}} \in (\mathbb{Q}^d)^+ \mid (\bar{\boldsymbol{v}}, 1) \models \varphi\}$ for the language of $\varphi$.

**Example 14.** *Consider sequences of the form $\boldsymbol{x}, A\boldsymbol{x}, A^2\boldsymbol{x}, \ldots, A^n\boldsymbol{x}$ such that $\boldsymbol{y}A^n\boldsymbol{x} = 0$ and $n \geq 0$ is minimal with this property where $\boldsymbol{y} \in \mathbb{Q}^{1 \times d}$ and $A \in \mathbb{Q}^{d \times d}$ are fixed and $\boldsymbol{x} \in \mathbb{Q}^d$. Theorem 3 implies that this language is accepted by a UHAT since it is defined by the $(LT)^2L$ formula $G[(\neg Last \to (\boldsymbol{y}\boldsymbol{x}_1 \neq 0 \wedge A\boldsymbol{x}_1 = \boldsymbol{x}_2)) \wedge (Last \to \boldsymbol{y}\boldsymbol{x}_1 = 0)]$, where $Last := \neg X\top$.*

**Example 15.** *Take the standard notion of 7-day Simple Moving Average (7-SMA); this can be generalized to larger sliding windows of 50-days, or 100 days, which are often used in finance. Using $(LT)^2L$, it is easy to show that the following notion of "uptrend" can be captured using UHAT: sequences of numbers such that the value at each time $t$ is greater than the 7-SMA value at time $t$. The formula for this is:*

$$G(X^7\top \to \varphi(x_1, \ldots, x_7))$$

*where $\varphi(\bar{x})$ is the formula $7x_7 > \sum_{i=1}^7 x_i$. Note here that $G\psi$ means (as usual for LTL) "globally" $\psi$, which can be written as $\neg(\top U\neg\psi)$. Similarly, $X^i$ means that $X$ is repeated $i$ times.*

We assume UHATs with positional encoding and a zero vector at the end of the input sequence (see Section 2). In the following we always assume that the components from the positional encoding are implicitly given and are not changed by any UHAT. So we write the sequence in Eq. (1) as $\boldsymbol{v}_1, \ldots, \boldsymbol{v}_n, \boldsymbol{0}$. We use the following results from [3] that also hold for UHATs over data sequences.

**Lemma 16.** *Let $d > 0$ and $\ell \in [1, d]$.*

1) *For every $b \in \{0, 1\}$ there is a UHAT with positional encoding that on every sequence $\boldsymbol{v}_1, \ldots, \boldsymbol{v}_n \in \mathbb{Q}^d$ with $v_i[\ell] \in \{0, 1\}$ for all $i \in [1, n]$ outputs the sequence $\boldsymbol{v}_1, \ldots, \boldsymbol{v}_{n-1}, (\boldsymbol{v}_n[1, \ell - 1], b, \boldsymbol{v}_n[\ell + 1, d])$.*

2) *There is a UHAT layer with positional encoding that on every sequence $\boldsymbol{v}_1, \ldots, \boldsymbol{v}_n \in \mathbb{Q}^d$ and for every $i \in [1, n - 1]$ picks attention vector $\boldsymbol{a}_i = \boldsymbol{v}_{i+1}$.*

3) *There is a UHAT layer with positional encoding that on every sequence $\boldsymbol{v}_1, \ldots, \boldsymbol{v}_n \in \mathbb{Q}^d$, for every $\ell \in [1, d]$ with $\boldsymbol{v}_1[\ell], \ldots, \boldsymbol{v}_{n-1}[\ell] \in \{0, 1\}$ and $\boldsymbol{v}_n[\ell] = 0$, and for every $i \in [1, n]$ picks attention vector $\boldsymbol{a}_i = \boldsymbol{v}_j$ with minimal $j \in [i, n]$ such that $\boldsymbol{v}_j[\ell] = 0$.*

Here, 2) and 3) directly follow from [3]. For 1) we remark that in [3] only the case $b = 0$ is shown. On input $\boldsymbol{v}_1, \ldots, \boldsymbol{v}_n$ as in 1), the UHAT uses positional encoding function $p(i, n) := (i, n)$

and a composition of affine transformations and ReLU to output at position $i \in [1, n]$ the vector $(\boldsymbol{v}_i[1, \ell-1], b_i, \boldsymbol{v}_i[\ell+1, d])$ where $b_i := \min\{\boldsymbol{v}_i[\ell], n-i\}$ if $b = 0$ and $b_i := \max\{\boldsymbol{v}_i[\ell], i-n+1\}$ if $b = 1$.

Using Lemma 16, we show that a UHAT can transform rational values $> 0$ to 1 and values $\leq 0$ to 0. This will be used to evaluate inequalities by outputting 1 for true and 0 for false.

**Lemma 17.** *Let $d > 0$ and $\ell \in [1, d]$. There is a UHAT with positional encoding that on every sequence $\boldsymbol{v}_1, \ldots, \boldsymbol{v}_{n+1} \in \mathbb{Q}^d$ outputs $\boldsymbol{v}_1', \ldots, \boldsymbol{v}_{n+1}' \in \mathbb{Q}^d$ with $\boldsymbol{v}_i' := (\boldsymbol{v}_i[1, \ell-1], b_i, \boldsymbol{v}_i[\ell+1, d])$ for all $i \in [1, n+1]$ where $b_i := 1$ if $\boldsymbol{v}_i[\ell] > 0$ and $b_i := 0$ otherwise.*

*Proof.* On input $\boldsymbol{v}_1, \ldots, \boldsymbol{v}_{n+1} \in \mathbb{Q}^d$, the first layer outputs at position $i \in [1, n+1]$ the vector $\boldsymbol{w}_i := (\boldsymbol{v}_i[1, \ell-1], r_i, \boldsymbol{v}_i[\ell+1, d])$ where $r_i := \max\{\boldsymbol{v}_i[\ell], 0\}$. Thus, $r_i = 0$ if $\boldsymbol{v}_i[\ell] \leq 0$ and $r_i > 0$ otherwise. The second layer turns the sequence $\boldsymbol{w}_1, \ldots, \boldsymbol{w}_{n+1}$ into $(0, \boldsymbol{w}_1), \ldots, (0, \boldsymbol{w}_{n+1})$. We then apply 1) of Lemma 16 to obtain the sequence $(0, \boldsymbol{w}_1), \ldots, (0, \boldsymbol{w}_n), (1, \boldsymbol{w}_{n+1})$, i.e. the last vector has first component 1, and all other vectors have first component 0. Let $\boldsymbol{u}_1, \ldots, \boldsymbol{u}_{n+1} \in \mathbb{Q}^{d+1}$ be the resulting sequence. The final layer uses attention score $\langle A\boldsymbol{u}_i, B\boldsymbol{u}_j \rangle$ for all $1 \leq i, j \leq n+1$ where the affine transformations $A, B \colon \mathbb{Q}^{d+1} \to \mathbb{Q}^{d+1}$ yield $A\boldsymbol{u}_i = (\boldsymbol{u}_i[\ell], 0, \ldots, 0)$ and $B\boldsymbol{u}_j = (\boldsymbol{u}_j[1], 0, \ldots, 0)$. Let $\boldsymbol{a}_i$ be the attention vector of position $i \in [1, n+1]$. Since $\boldsymbol{u}_i[\ell] \geq 0$, we have $\boldsymbol{a}_i[1] = 0$ if $\boldsymbol{u}_i[\ell] = 0$ and $\boldsymbol{a}_i[1] = 1$ if $\boldsymbol{u}_i[\ell] > 0$. The layer outputs $\boldsymbol{v}_i' := (\boldsymbol{u}_i[2, \ell], \boldsymbol{a}_i[1], \boldsymbol{u}_i[\ell+2, d+1])$ at position $i \in [1, n+1]$. $\square$

We now prove Theorem 3. We claim that for every $(\text{LT})^2\text{L}$ formula $\varphi$ of dimension $d$ and every $m \geq d$ there exists a UHAT $T_{\varphi,m}$ with positional encoding that on every sequence $\boldsymbol{w}_1, \ldots, \boldsymbol{w}_n, \boldsymbol{0} \in \mathbb{Q}^m$ outputs a sequence $\boldsymbol{w}_1', \ldots, \boldsymbol{w}_n', \boldsymbol{0} \in \mathbb{Q}^{m+1}$ such that for all $i \in [1, n]$ we have $\boldsymbol{w}_i'[1, m] = \boldsymbol{w}_i$ and $\boldsymbol{w}_i'[m+1] = 1$ if $(\bar{\boldsymbol{v}}, i) \models \varphi$ and $\boldsymbol{w}_i'[m+1] = 0$ otherwise, where $\bar{\boldsymbol{v}} := (\boldsymbol{w}_1[1, d], \ldots, \boldsymbol{w}_n[1, d])$. Then the theorem follows since for every $(\text{LT})^2\text{L}$ formula $\varphi$ of dimension $d$ the UHAT $T_{\varphi,d}$ outputs on every sequence $\bar{\boldsymbol{v}} = (\boldsymbol{v}_1, \ldots, \boldsymbol{v}_n)$ of vectors in $\mathbb{Q}^d$ extended with the vector $\boldsymbol{0} \in \mathbb{Q}^d$ a sequence $\boldsymbol{v}_1', \ldots, \boldsymbol{v}_n', \boldsymbol{0} \in \mathbb{Q}^{d+1}$ such that $\boldsymbol{v}_1'[d+1] > 0$ if and only if $(\bar{\boldsymbol{v}}, 1) \models \varphi$. Thus, $T_{\varphi,d}$ accepts $L(\varphi)$ by taking the acceptance vector $\boldsymbol{t} := (0, \ldots, 0, 1) \in \mathbb{Q}^{d+1}$.

We prove the claim by induction on the structure of $(\text{LT})^2\text{L}$ formulas. If the formula is a unary numerical predicate $\Theta$, then we can use the positional encoding $p(i, n+1) := \theta_n(i)$ for all $i \in [1, n]$ and $p(n+1, n+1) := 0$ to output on every sequence $\boldsymbol{w}_1, \ldots, \boldsymbol{w}_n, \boldsymbol{0} \in \mathbb{Q}^m$ the sequence $(\boldsymbol{w}_1, p(1, n+1)), \ldots, (\boldsymbol{w}_n, p(n, n+1)), (\boldsymbol{0}, p(n+1, n+1)) \in \mathbb{Q}^{m+1}$.

If the formula is an atom $\psi_k(\boldsymbol{x}_1, \ldots, \boldsymbol{x}_{k+1})$ of the form $\langle \boldsymbol{a}, (\boldsymbol{x}_1, \ldots, \boldsymbol{x}_{k+1}) \rangle + b > 0$, the UHAT $T_{\psi_k,m}$ adds in its first layer a component that is set to 1 to the top of every vector, outputting on every sequence $\boldsymbol{w}_1, \ldots, \boldsymbol{w}_n, \boldsymbol{0} \in \mathbb{Q}^m$ the sequence $(1, \boldsymbol{w}_1), \ldots, (1, \boldsymbol{w}_n), (1, \boldsymbol{0})$. Then we apply 1) of Lemma 16 to turn this sequence into $(1, \boldsymbol{w}_1), \ldots, (1, \boldsymbol{w}_n), \boldsymbol{0}$. Next, $T_{\psi_k,m}$ uses $k$ layers to allow each position to gather the first $d+1$ components of its $k$ right neighbors. More precisely, the $\ell$-th layer, for $\ell \in [1, k]$, on sequence $\boldsymbol{u}_1, \ldots, \boldsymbol{u}_{n+1}$ uses 2) of Lemma 16 to get for every position $i \in [1, n]$ the attention vector $\boldsymbol{a}_i = \boldsymbol{u}_{i+1}$ and the attention vector $\boldsymbol{a}_{n+1}$ is arbitrary. Note that if $\ell = 1$, then $\boldsymbol{a}_n = \boldsymbol{0}$. Then it applies an affine transformation to output at position $i \in [1, n]$ the vector $(\boldsymbol{a}_i[1, d+1], \boldsymbol{u}_i)$ and using 1) at position $n+1$ the vector $(0, \boldsymbol{a}_{n+1}[2, d+1], \boldsymbol{u}_{n+1})$. Let $\boldsymbol{u}_1, \ldots, \boldsymbol{u}_{n+1} \in \mathbb{Q}^{k(d+1)+m+1}$ be the output of the $k$-th of those layers. We add another layer that using a composition of ReLU and affine functions outputs at every position $i \in [1, n+1]$ the vector $\boldsymbol{u}_i' := (\boldsymbol{u}_i[k(d+1)+2, k(d+1)+m+1], r_i) \in \mathbb{Q}^{m+1}$ where $r_i := \min\{\boldsymbol{u}_i[1], \langle \boldsymbol{a}, \hat{\boldsymbol{u}}_i \rangle + b\}$ and

$$\hat{\boldsymbol{u}}_i := (\boldsymbol{u}_i[2, d+1], \boldsymbol{u}_i[d+3, 2(d+1)], \ldots, \boldsymbol{u}_i[kd+k+2, (k+1)(d+1)]),$$

which contains the first $d$ components of the initial input vector and its $k$ right neighbors. That is, for all $i \in [1, n]$ we have that $r_i = 0$ if $\langle \boldsymbol{a}, \hat{\boldsymbol{u}}_i \rangle + b \leq 0$ or $i + k > n$ since $\boldsymbol{u}_i[1]$ can only be 0 if it was gathered from the vector at position $n+1$ using attention. Furthermore, $r_i > 0$ if $\langle \boldsymbol{a}, \hat{\boldsymbol{u}}_i \rangle + b > 0$ and $i + k \leq n$. Note that $\boldsymbol{u}_i'[1, m]$ is equal to the input vector $\boldsymbol{w}_i$ from the beginning if $i \in [1, n]$ and $\boldsymbol{0}$ if $i = n+1$. Finally, we apply Lemma 17 followed by 1) to output at position $i \in [1, n]$ the vector $\boldsymbol{w}_i' := (\boldsymbol{u}_i'[1, m], r_i')$ where $r_i' := 1$ if $r_i > 0$ and $r_i' := 0$ otherwise and at position $n+1$ the vector $\boldsymbol{w}_{n+1}' := (\boldsymbol{u}_{n+1}'[1, m], 0) = \boldsymbol{0}$. Thus, for all $i \in [1, n]$ we have that $\boldsymbol{w}_i'[m+1] = 1$ if $(\bar{\boldsymbol{v}}, i) \models \psi_k$ and $\boldsymbol{w}_i'[m+1] = 0$ otherwise, where $\bar{\boldsymbol{v}} := (\boldsymbol{w}_1[1, d], \ldots, \boldsymbol{w}_n[1, d])$.

Let us now continue with the inductive step where we assume that $\varphi$ and $\psi$ are $(\text{LT})^2\text{L}$ formulas of dimension $d$ such that for all $m \geq d$ we already showed existence of the UHATs $T_{\varphi,m}$ and $T_{\psi,m+1}$.

For the cases $\neg\varphi$, $\varphi\vee\psi$, and $X\varphi$ we refer to Appendix D. For $\varphi U\psi$ define the UHAT $T_{\varphi U\psi,m}$ that first applies $T_{\psi,m+1} \circ T_{\varphi,m}$ outputting a sequence $\boldsymbol{u}_1,\ldots,\boldsymbol{u}_n,\boldsymbol{0} \in \mathbb{Q}^{m+2}$. Observe that $(\bar{\boldsymbol{v}},i) \models \varphi U\psi$ for $i \in [1,n]$ and $\bar{\boldsymbol{v}} := (\boldsymbol{u}_1[1,d],\ldots,\boldsymbol{u}_n[1,d])$ if and only if for the minimal $j \in [i,n]$ with $(\bar{\boldsymbol{v}},j) \models \neg\varphi\vee\psi$ we have $(\bar{\boldsymbol{v}},j) \models \psi$ and such a $j$ exists. Equivalently, for the minimal $j \in [i,n+1]$ with $\min\{\boldsymbol{u}_j[m+1], 1-\boldsymbol{u}_j[m+2]\} = 0$ it holds that $\boldsymbol{u}_j[m+2] = 1$. To check this, we first add a layer that outputs at position $i \in [1,n+1]$ the vector $\boldsymbol{u}'_i := (\boldsymbol{u}_i, \min\{\boldsymbol{u}_i[m+1], 1-\boldsymbol{u}_i[m+2]\}) \in \mathbb{Q}^{m+3}$. Finally, we add a layer that uses 3) of Lemma 16 to get attention vector $\boldsymbol{a}_i = \boldsymbol{u}'_j$ with $j \in [i,n+1]$ minimal such that $\boldsymbol{u}'_j[m+3] = 0$. The layer then outputs at position $i \in [1,n+1]$ the vector $\boldsymbol{w}'_i := (\boldsymbol{u}'_i[1,m], \boldsymbol{a}_i[m+2]) \in \mathbb{Q}^{m+1}$.

## 6    Concluding remarks

We initiated the study of the expressive power of transformers, when the input is a sequence of (tuples of) numbers, which is the setting for applications like time series analysis/forecasting. Our results indicate an increased expressiveness of transformers on such input data, in comparison to the previous formal language theoretic setting (see survey [40]), i.e., when a token embedding function (with a bounded number of tokens) is first applied before feeding the input to a transformer. More precisely, this represents for Unique Hard Attention Transformers (UHAT) a jump from the complexity class $AC^0$ to $TC^0$ (since $AC^0 \subsetneq TC^0$), and the jump from regular to non-regular languages (when position encoding is not allowed). On the positive side, we successfully developed an expressive class of logical languages recognized by UHAT in terms of a logic called locally testable LTL, which extends previously identified logic for UHAT for strings over finite alphabets [2, 3].

**Limitations.** While we follow the standard formalization of transformer encoders in Formal Languages and Neural Networks (e.g. [3, 20, 21, 40]), limitations of the models are known (see [40] for a thorough discussion). For example, used real numbers could be of *unbounded* precision, which allow one to precisely represent values of $\sin$ and $\cos$ functions (actually used in practice for positional encoding). In addition, the positional encoding used by the model could be *uncomputable*. Three answers can be given. First, an *upper bound complexity* on the model with unbounded precision and arbitrary positional encodings (e.g. in $TC^0$) still applies in the case of bounded precision. Second, limiting the power of UHAT (e.g. allow only rational numbers, and assuming efficient (i.e. uniform $TC^0$) computability of the positional encoding $p : \mathbb{N} \times \mathbb{N} \to \mathbb{Q}^d$), our proof in fact yields *uniformity* of our $TC^0$ upper bound. Third, our lower bound for non-regularity of UHAT (cf. Theorem 2) holds even with only rational numbers and no positional encodings. Finally, to alleviate these issues, we have always made an explicit distinction between UHAT with and without positional encodings.

**Future directions.** Our paper opens up a plethora of research avenues on the expressive power of transformers on data sequences. In particular, one could consider other transformer encoder models that have been considered in the formal language theoretic setting to transformers (see the survey [40]). For example, instead of unique hard attention mechanism, we could consider the expressive power of transformers on data sequences using *average hard attention* mechanism. Similar question could be asked if we use a softmax function instead of a hard attention, which begs the question of which numerical functions could be computed in different circuit complexity classes like $TC^0$. Another important question concerns a logical characterization for UHAT over sequences of numbers. This is actually still an open question even for the case of finite alphabets. Barcelo et al. [3] showed that first-order logic (equivalently, LTL) with monadic numerical predicates (called LTL(Mon)) is subsumed in UHAT with arbitrary position encodings, i.e., the transformer model that we are generalizing in this paper to sequences of numbers. There are UHAT languages (e.g. the set of palindromes) that are not captured by this. As remarked in [3], LTL(Mon) can be extended with arbitrary linear orders on the positions (parameterized by lengths), which then can define the palindromes. [An analogous extension for $(LT)^2L$ can define palindromes over an infinite alphabet.] However, it is possible to show that the resulting logic is still not expressive enough to capture the full generality of UHAT. That said, although our logic does not capture the full UHAT, it can still be used to see at a glance what languages can be recognized by UHAT (e.g. Simple Moving Averages). There could perhaps be a hope of obtaining a precise logical characterization if we restrict the model of UHAT. The recent paper [2] showed that LTL(Mon) captures precisely the languages of masked UHAT with position encodings with "finite image". It is interesting to study similar restrictions for UHAT in the case of sequences of numbers.

**Acknowledgments**     Funded by the European Union (ERC, LASD, 101089343 and FIN-ABIS, 101077902). Views and opinions expressed are however those of the authors only and do not necessarily reflect those of the European Union or the European Research Council Executive Agency. Neither the European Union nor the granting authority can be held responsible for them.

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

# A    Example UHAT with real parameters

We present here an example UHAT in which two real numbers $\alpha$ and $\beta$ occur (each occurs once in a matrix associated to a particular layer) such that the simple sequence

$$(1, 0, \boldsymbol{e}_1)(1, 0, \boldsymbol{e}_2)(1, 0, \boldsymbol{e}_3), \tag{2}$$

where $\boldsymbol{e}_i \in \mathbb{R}^3$ is the $i$-th unit vector, is accepted if and only if $\alpha\beta = 2$ and $\alpha = \beta$. This shows that even if we restrict the input sequence to a particular number $B$ of bits (e.g. the number of bits to represent the input sequence Eq. (2)), it is not possible to replace the real constants $\alpha$ and $\beta$ by rational numbers without changing the accepted sequences of up to $B$ bits: The sequence Eq. (2) is only accepted if $\alpha = \beta = \sqrt{2}$. And if we change $\alpha$ or $\beta$ in any way, the sequence Eq. (2) will not be accepted anymore.

1. Input layer:
$$(1, 0, \boldsymbol{e}_1)(1, 0, \boldsymbol{e}_2)(1, 0, \boldsymbol{e}_3).$$

2. Using a standard encoding layer, we multiply the first component in each position by $\alpha$. Result:
$$(\alpha, 0, \boldsymbol{e}_1)(\alpha, 0, \boldsymbol{e}_2)(\alpha, 0, \boldsymbol{e}_3).$$

3. Using attention, we can apply distinct affine transformations to the first three positions. We choose the following affine transformations. The first position is unchanged. The second position is mapped to $(1, 0, \boldsymbol{e}_2)$, and the third position is mapped to $(0, \alpha, \boldsymbol{e}_3)$, using a matrix that flips the first and second component. Result:
$$(\alpha, 0, \boldsymbol{e}_1)(1, 0, \boldsymbol{e}_2)(0, \alpha, \boldsymbol{e}_3).$$

4. Using a standard encoding layer, we multiply the first component in each position with $\beta$. Result:
$$(\alpha\beta, 0, \boldsymbol{e}_1)(\beta, 0, \boldsymbol{e}_2)(0, \alpha, \boldsymbol{e}_3).$$

5. Finally, by using our result on $(\mathrm{LT})^2\mathrm{L}$, we can build further layers so that we accept if and only if (i) the first component of the first position equals 2 and (ii) the first component of the second position equals the second component of the third position. Thus, we accept our original input if and only if $\alpha\beta = 2$ and $\alpha = \beta$.

# B    Omitted definitions and proofs in Section 3

## B.1    Descriptional size

Let $x \in \mathbb{R}$ be some real number. The *(descriptional) size* of $x$ is $\mathrm{size}(x) = 1 + \lceil \log_2(|p| + 1) \rceil + \lceil \log_2(|q| + 1) \rceil$ if $x = \frac{p}{q}$ is a rational number (where $p$ and $q$ are relatively prime integers) and $\mathrm{size}(x) = 1$ if $x$ is an irrational number. Note that in the latter case we use the number 1 as some placeholder, since we do not have to represent irrational numbers in any algorithm. However, for analysis of the sizes in our constructions we still need some value.

Let $\boldsymbol{v} \in \mathbb{R}^d$ be a vector. The *size* of $\boldsymbol{v}$ is $\mathrm{size}(\boldsymbol{v}) = n + \sum_{i=1}^d \mathrm{size}(v_i)$ where $\boldsymbol{v} = (v_1, \ldots, v_d)^T$.

Let $M \in \mathbb{R}^{m \times n}$ be a matrix. The *size* of $M$ is $\mathrm{size}(M) = mn + \sum_{1 \leq i \leq m, 1 \leq j \leq n} \mathrm{size}(a_{ij})$ where $M = (a_{ij})_{1 \leq i \leq m, 1 \leq j \leq n}$.

Let $A \colon \mathbb{R}^d \to \mathbb{R}^e$ be an affine transformation, i.e., we have $A(\boldsymbol{x}) = B\boldsymbol{x} + \boldsymbol{c}$ with $B \in \mathbb{R}^{d \times e}$ and $\boldsymbol{c} \in \mathbb{R}^e$. Then the *size* of $A$ is $\mathrm{size}(A) = \mathrm{size}(B) + \mathrm{size}(c) + 1$.

Now, let $p \in \mathbb{R}[X_1, \ldots, X_n]$ be a polynomial. Then we have $p(X_1, \ldots, X_n) = \sum_{0 \leq r_1, \ldots, r_n \leq k} c_{r_1, \ldots, r_n} X_1^{r_1} \cdots X_n^{r_n}$ for some numbers $k \in \mathbb{N}$ and $c_{r_1, \ldots, r_n} \in \mathbb{R}$. The *size* of $p$ is

$$\mathrm{size}(p) = \sum_{0 \leq r_1, \ldots, r_n \leq k} \mathrm{size}(c_{r_1, \ldots, r_n}) + \mathrm{size}(r_1) + \cdots + \mathrm{size}(r_n) + n \,.$$

Let $\alpha$ be a polynomial constraint. We define the *size* of $\alpha$ inductively on the structure of the formula as follows:

- if $\alpha = (p(X_1, \ldots, X_n) \sim 0)$ with $\sim \in \{>, \geq\}$ is an atom. Then $\text{size}(\alpha) = \text{size}(p) + 1$.
- if $\alpha = \bigwedge_{1 \leq i \leq k} \beta_i$ or $\alpha = \bigvee_{1 \leq i \leq k} \beta_i$ is a formula with PCs $\beta_1, \ldots, \beta_k$, then $\text{size}(\alpha) = k + \sum_{1 \leq i \leq k} \text{size}(()\beta_i)$.

Let $R = (\phi_1, D_1), \ldots, (\phi_k, D_k)$ be a CPR. Then the *size* of this CPR is $\text{size}(R) = k + \sum_{i=1}^{k} \text{size}(\phi_i) + \text{size}(D_k)$.

## B.2 ReLU case in Lemma 6

Let us now consider a ReLU layer. Assume that we compute the ReLU-value for the $j$-th component, i.e., we compute $\max\{0, x_{i,j}\}$ for each $i \in \{1, \ldots, n+1\}$ where $x_{i,j}$ is the $j$-th component of $x_i$. From each conditional assignment $\varphi_{i,k} \to D_{i,k}$ with $i \in \{1, \ldots, n+1\}$ and $1 \leq k \leq s_i$ we construct two new conditional assignments:

1. $\langle e_{(i-1) \cdot n + j}, D_{i,k} \bar{x} \rangle \geq 0 \to D_{i,k}$ where $e_h$ is the $h$-th unit vector.

2. $\langle -e_{(i-1) \cdot n + j}, D_{i,k} \bar{x} \rangle > 0 \to M D_{i,k}$ where $M = (m_{gh})_{1 \leq g, h \leq d' \cdot (n+1)}$ is the matrix with $m_{gh} = 1$ if $g = h \neq (i-1) \cdot n + j$ and $m_{gh} = 0$ otherwise (i.e., $M$ is the unit matrix except for the $(i-1) \cdot j$-th entry).

Now, if the $j$-th component of $x_i$ is non-negative, only the first conditional assignment is satisfied and the value of this component is left untouched. Otherwise, the $j$-th component is negative. But then the second conditional assignment is satisfied and the value of this component is set to $0$ (while the others stay unchanged). So, we obtain again some polynomial sized CPR with the same number of alternations as before.

## B.3 Omitted proofs

*Proof of Proposition 5.* Let $t \in \mathbb{R}^e$ be the acceptance criterion of the UHAT and $f \colon \mathbb{R}^{d \cdot (n+1)} \to \mathbb{R}^{e \cdot (n+1)}$ be the computed function for inputs of length $n$. By Lemma 6, we can construct in polynomial time an $O(\ell)$-alternation-bounded CPR computing $f$. So, let $\varphi_{i,k} \to D_{i,k}$ be the conditional assignments in this CPR (for $1 \leq i \leq n+1$ and $1 \leq k \leq s_i$). Then we obtain a PC from this CPR as follows:

$$\bigvee_{J=1}^{s_1} \varphi_{1,J} \wedge \langle t, D_{1,J} \bar{x} \rangle > 0 \,.$$

Note that this PC has still polynomial size, accepts an input sequence $(x_1, \ldots, x_n, 0)$ if, and only if, the UHAT accepts $(x_1, \ldots, x_n)$, and — if the CPR is $a$-alternation-bounded — then it has at most $a + 2$ alternations of disjunctions and conjunctions. $\square$

*Proof of Proposition 7.* Consider a constraint $p(X_1, \ldots, X_n) > 0$ (or $\geq 0$ resp.) in $\alpha$. Then $p \in \mathbb{R}[X_1, \ldots, X_n]$ is a polynomial of degree at most 2, i.e., there are real numbers $c_{i,j,r,s} \in \mathbb{R}$ such that

$$p(X_1, \ldots, X_n) = \sum_{0 \leq r+s \leq 2} \sum_{1 \leq i \leq j \leq n} c_{i,j,r,s} X_i^r X_j^s \,.$$

Now, construct two vectors $u$ and $w$ with a component for each tuple $(i, j, r, s)$: $u_{i,j,r,s} = c_{i,j,r,s}$ and $w = i, j, r, s = X_i^r X_j^s$. Then it is clear that $p(X_1, \ldots, X_n) = \langle u, w(X_1, \ldots, X_n) \rangle$ holds.

Application of Lemma 8 yields a vector $v \in \mathbb{Q}^t$ with $\|v\|_2 \leq (2mt)^{O(1)}$ and $u \sim_{2m} v$. Note that we need to consider rational numbers up to size $2m$ due to the fact that substitution of the variables in $w$ by rational numbers $x \in \mathbb{Q}^n_{\leq m}$ yields a rational vector $w(x) \in Q^t_{\leq 2m}$. Let $p'(X_1, \ldots, X_n)$ be the polynomial obtained from $p$ be replacing the coefficients $c_{i,j,r,s} \in \mathbb{R}$ by $v_{i,j,r,s} \in \mathbb{Q}$. Then for each $x \in \mathbb{Q}^n_{\leq m}$ we have $p(x) = \langle u, w(x) \rangle > 0$ (resp. $\geq 0$) if, and only if, $p'(x) = \langle v, w(x) \rangle > 0$ (resp. $\geq 0$).

Replacing each real polynomial $p$ in $\alpha$ by the constructed rational polynomial $p'$ results in a rational PC $\alpha'$ with $[\![\alpha]\!] \cap \mathbb{Q}^n_{\leq m} = [\![\alpha]\!] \cap \mathbb{Q}^n_{\leq m}$. $\square$

## B.4 Proof of Lemma 10

The rest of this subsection is devoted to proving Lemma 8, for which we rely on results from convex geometry, which requires some terminology. For a set $S \subseteq \mathbb{R}^n$, we define the *convex hull of $S$* as

$$\text{conv.hull } S = \{\lambda_1 s_1 + \cdots + \lambda_m s_m \mid m > 0, \ s_1, \ldots, s_m \in S,$$
$$\lambda_1, \ldots, \lambda_m \in [0, 1], \ \lambda_1 + \cdots + \lambda_m = 1\}$$

and the *cone generated by $S$* as

$$\text{cone } S = \{\lambda_1 s_1 + \cdots + \lambda_m s_m \mid m > 0, \ s_1, \ldots, s_m \in S, \ \lambda_1, \ldots, \lambda_m \geq 0\}.$$

We will also rely on Carathéodory's theorem, which says that a points in cones and convex sets can be obtained from at most $n$ points. See, for example, [35, Theorems 7.1i and 7.1j].

**Theorem 18** (Carathéodory's Theorem). *Let $S \subseteq \mathbb{R}^n$. For every $x \in$ conv.hull $S$, there are $x_1, \ldots, x_n \in S$ with $x \in$ conv.hull $\{x_1, \ldots, x_n\}$. Moreover, for every $y \in$ cone $S$, there are $y_1, \ldots, y_n \in S$ with $y \in$ cone $\{y_1, \ldots, y_n\}$.*

A *polyhedron* is a set of the form $\{x \in \mathbb{R}^n \mid Ax \geq b\}$, where $A \in \mathbb{R}^{m \times n}$ and $b \in \mathbb{R}^m$ for some $m \in \mathbb{N}$. If the matrix $A$ and the vector $b$ are rational, then the polyhedron is called a *rational polyhedron*. It is a standard result about polyhedra that if $A$ and $b$ are rational of size at most $m$, then the polyhedron $P = \{x \in \mathbb{R}^n \mid Ax \geq b\}$ can be written as

$$P = \text{conv.hull } \{x_1, \ldots, x_s\} + \text{cone } \{y_1, \ldots, y_t\},$$

with rational vectors $x_1, \ldots, x_s, y_1, \ldots, y_t$, where each vector has size polynomial in $mn$. See, for example, [35, Theorem 10.2].

*Proof.* We define the matrix $B \in \mathbb{Q}^{(k+\ell) \times n}$ and the vector $b \in \mathbb{Q}^{k+\ell}$ as

$$B = \begin{bmatrix} A \\ A' \end{bmatrix} \qquad\qquad b = \begin{bmatrix} z \\ z' \end{bmatrix}.$$

and consider the polyhedron $P = \{x \in \mathbb{R}^n \mid Bx \geq b\}$. As mentioned above, we can write

$$P = \text{conv.hull } \{x_1, \ldots, x_s\} + \text{cone } \{y_1, \ldots, y_t\},$$

where $x_1, \ldots, x_s, y_1, \ldots, y_t$ are rational vectors of size polynomial in $mn$. By our assumption, there exists an $s \in \mathbb{R}^n$ with $As \gg z$ and $A's \geq z'$. By Carathéodory's Theorem, wlog, $s$ belongs to the smaller polyhedron

$$Q = \text{conv.hull } \{x_1, \ldots, x_n\} + \text{cone } \{y_1, \ldots, y_n\}.$$

Now note that we have $Au \geq z$ and $A'u \geq z'$ for every $u \in U := \{x_1, \ldots, x_n, y_1, \ldots, y_n\}$. We claim that the vector

$$r = \tfrac{1}{n}(x_1 + \cdots + x_n) + y_1 + \cdots + y_n$$

satisfies $Ar \gg z$ and $A'r \geq z'$. Indeed, it clearly belongs to $Q \subseteq P$ and thus satisfies $A'r \geq z'$. Moreover, for every row $a^\top x > z$ of $Ax \gg z$, there must be a $u \in U$ with $a^\top u > z$—otherwise, we would would have $a^\top u = z$ for every $u \in U$ and thus $a^\top s = z$. In particular, we have $Ar \gg z$. Finally, the vector $r$ has size at most $\|x_1\|_2 + \cdots + \|x_n\|_2 + \|y_1\|_2 + \cdots + \|y_n\|_2$, which is polynomial in $mn$, since each $x_i$ and each $y_i$ has size polynomial in $mn$. $\square$

## B.5 Proof of Lemma 8

*Proof of Lemma 8.* Collect the set of all inequalities $\langle w, u \rangle > z$ or $\langle w, u \rangle \geq z$ with $w \in \mathbb{Q}^t_{\leq m}$ and $z \in \mathbb{Q}_{\leq m}$ that are satisfied for $u$. This results in two large matrices $A \in \mathbb{Q}^{k \times t}_{\leq m}$ and $A' \in \mathbb{Q}^{\ell \times t}_{\leq m}$ and vectors $z \in \mathbb{Q}^k_{\leq m}$ and $z' \in \mathbb{Q}^\ell_{\leq m}$ such that we have $u \sim_m v$ if and only if $Av \gg z$ and $A'v \geq z'$. Thus, we can construct $c'$ using Lemma 10. Observe that the bound from Lemma 10 does not depend on the (exponentially large) $k$ and $\ell$. $\square$

*Proof of Proposition 11.* Let $T$ be some UHAT with positional encoding and $n, m \in \mathbb{N}$ be two natural numbers. In the following, we consider input sequences of $T$ having the length $n$ and size $m$. By Propositions 5 and 7 there is polynomial sized, $O(\ell)$-alternation-bounded, and rational PC $\alpha$ (where $\ell$ is the number of layers in $T$) such that the set of sequences of size $m$ accepted by the UHAT $T$ equals $[\![\alpha]\!] \cap \mathbb{Q}_{\leq m}^{n \cdot d}$.

We finally show that the PC $\alpha$ can be realized as a circuit of constant depth and polynomial size. So, consider a constraint of the form $p(\bar{x}) \sim 0$ where $\sim \in \{\geq, >\}$, $p$ is a polynomial of degree at most 2 and $\bar{x}$ represents the input sequence. Since addition and multiplication of rational numbers are realizable in $TC^0$ [7], it is clear that the computation of the value $p(\bar{x})$ is also realizable. Additionally, checking whether this value is $\geq 0$ (or $> 0$, resp.) is a simple check of the bit representing the signum (and checking that the numerator has at least one non-zero bit).

Finally, we have to connect all the atoms of the form $p(\bar{x}) \sim 0$ to the Boolean formula $\alpha$. Since $\alpha$ alternates only a bounded number of times between disjunctions and conjunctions, we can realize the complete formula $\alpha$ in a circuit of constant depth and with polynomial size. Since $\alpha$ is equivalent (up to the input size $m$), the UHAT $T$ is realizable in $TC^0$. Note that here, we need to perform iterated addition of rational numbers, which requires iterated multiplication of integers represented in binary. The latter is well-known to be possible in $TC^0$ [33, 34] (even uniformly [22], but this is not needed in our setting). $\qquad\square$

## C   Proof of non-regularity in Section 4

Recall that

$$\text{Double} := \{(r_1, \ldots, r_n) \in \mathbb{Q}^n \mid n \geq 1 \text{ and } 2r_i < r_{i+1} \text{ for all } 1 \leq i < n\}.$$

We next define the notion of symbolic automata [10, 11, 43]. A *symbolic automaton* is a tuple $(Q, \delta, q_0, F)$, where $Q$ is a finite set of states, $q_0 \in Q$ is an initial state, $F \subseteq Q$ is a set of final states, and $\delta$ is a set of transition rules of the form $(p, S, q)$, where $S \subseteq \mathbb{Q}$. For $a \in \mathbb{Q}$, we write $p \to_a q$ (read "there is a transition from $p$ to $q$ reading $a$) if there is a transition rule $(p, S, q)$ such that $a \in S$. Slightly abusing notation, for a set $S \subseteq \mathbb{Q}$, we also write $p \to_S q$ to mean that $(p, S, q)$ is a transition rule in $\delta$. The notion of a run, and an accepting run can then be defined in exactly the same way as for finite automata (e.g. see [37]); namely, it is a sequence of transitions $q_0 \to_{a_1} \cdots \to_{a_n} q_n$, where $q_n \in F$, reading the sequence $w = a_1 \cdots a_n$ over $\mathbb{Q}$.

To prove that there is no symbolic automaton recognizing Double, let us assume to the contrary that such an automaton $A$ exists, say, with $n$ states. Consider a sufficiently long $w = a_1 \cdots a_m \in$ Double (i.e. of length at least $n$), and an accepting run of $A$:

$$q_0 \to_{S_1} \cdots \to_{S_m} q_m$$

where each $a_i \in S_i$ and $q_m \in F$. Since $m + 1 > n$, by pigeonhole principle, it must be the case that $q_r = q_s$ for some $r < s$. Thus, also the sequence

$$a_1 \cdots a_r (a_{r+1} \cdots a_s)^2 a_{s+1} \cdots a_m$$

is accepted by $A$ and therefore contained in Double. This is a contradiction since Double imposes $a_s < a_s$.

## D   Omitted cases in proof of Theorem 3

For $\neg\varphi$ the UHAT $T_{\neg\varphi,m}$ first applies $T_{\varphi,m}$ and on the obtained sequence $\boldsymbol{u}_1, \ldots, \boldsymbol{u}_n, \boldsymbol{0} \in \mathbb{Q}^{m+1}$ uses a further layer followed by 1) to output $\boldsymbol{w}_i' := (\boldsymbol{u}_i[1, m], 1 - \boldsymbol{u}_i[m+1])$ at position $i \in [1, n]$ and $\boldsymbol{0} \in \mathbb{Q}^{m+1}$ at position $n + 1$.

For $\varphi \vee \psi$ we define the UHAT $T_{\varphi \vee \psi, m}$ that first applies $T_{\psi, m+1} \circ T_{\varphi, m}$ followed by a layer that on sequence $\boldsymbol{u}_1, \ldots, \boldsymbol{u}_{n+1} \in \mathbb{Q}^{m+2}$ outputs $\boldsymbol{w}_1', \ldots, \boldsymbol{w}_{n+1}' \in \mathbb{Q}^{m+1}$ with

$$\boldsymbol{w}_i' := (\boldsymbol{u}_i[1, m], \max\{\boldsymbol{u}_i[m+1], \boldsymbol{u}_i[m+2]\})$$

for all $i \in [1, n+1]$. Note that if $\boldsymbol{u}_{n+1} = \boldsymbol{0}$, then also $\boldsymbol{w}_{n+1}' = \boldsymbol{0}$.

For $X\varphi$ the UHAT $T_{X\varphi,m}$ first applies $T_{\varphi,m}$ to output a sequence $\boldsymbol{u}_1,\ldots,\boldsymbol{u}_{n+1} \in \mathbb{Q}^{m+1}$ with $\boldsymbol{u}_{n+1} = \boldsymbol{0}$. With an additional layer that uses 2) to get attention vector $\boldsymbol{a}_i = \boldsymbol{u}_{i+1}$ for all $i \in [1,n]$ it then outputs at position $i \in [1,n]$ the vector $\boldsymbol{w}'_i := (\boldsymbol{u}_i[1,m], \boldsymbol{a}_i[m+1])$ and at position $n+1$ the vector $\boldsymbol{w}'_{n+1} := (\boldsymbol{u}_{n+1}[1,m], 0)$ after applying 1).

