# OpenReview forum: "The Power of Hard Attention Transformers on Data Sequences: A formal language theoretic perspective"
_NeurIPS.cc/2024/Conference — NeurIPS 2024 poster_

### Official Review · Reviewer_TvTa · 2024-07-12

**Soundness:** 3
**Presentation:** 3
**Contribution:** 2
**Rating:** 6
**Confidence:** 4

**Summary:**

This paper focuses on analyzing the expressive power of transformers through the lens of formal languages, inspired by Angluin's approach. Specifically, while traditional unique hard attention transformers (UHAT) on strings are associated with $AC^0$ and regular languages definable in first-order logic, the authors here explore the implications of using data sequences instead of words as input. The main findings are:
- UHATs over data sequences with positional encodings fall within $TC^0$, not $AC^0$.
- Non-regular languages can be recognized even by masked UHATs without positional encodings.
- UHATs can recognize all languages definable in $(LT^2)L$, an extension of LTL with unary predicates and local arithmetic tests over finite windows over the data.

**Strengths:**

**S1: Important Contribution**
Understanding the expressive power of significant classes of transformers is crucial, given their widespread popularity. Moreover, considering data sequences is a natural and worthwhile extension. The results obtained are both elegant and insightful.

**S2: Clarity**
For readers familiar with the topic, this is a well-written paper. The authors effectively set the stage and explain most concepts and proof ideas in sufficient detail within the main text.

**S3: Elegant Proofs**
The results linking UHAT with $TC^0$ involve a well-crafted stepwise reduction using polynomial equations and polyhedral analysis.

**Weaknesses:**

**W1: Coverage of $TC^0$**
While the characterization of UHAT on data sequences in $TC^0$ is commendable, it does not fully encompass $TC^0$. A precise characterization of the power of UHATs would strengthen the paper. This observation also applies to the other main results: Which logic precisely captures UHAT?

**W2: Accessibility**
The authors make little effort to make the paper accessible to the broader ML community. As it stands, the paper could have been submitted to any computer science theory conference. Consequently, the results may not be surprising to those communities.

**W3: Practical Implications**
The paper lacks discussion on the practical implications of the theoretical analysis for transformer design. Coupling the theory with practical insights would enhance the paper's value.

In summary, the paper provides significant insights into the expressive power of transformers over data sequences, presenting elegant and theoretically sound results. However, a broader characterization of UHATs' power, increased accessibility for a wider audience, and practical implications would strengthen the paper further.

**Questions:**

Please comment on **W1** and **W3**.

**Limitations:**

This has been addressed in a satisfactory way by the authors.

---

> ### Author Rebuttal · Authors · 2024-08-07
>
> **Q: How much of $TC^0$ is covered?**
>
> A: Firstly, Barcelo et al. [3] showed that there is an $AC^0$ language over the alphabet {0,1} that is not in UHAT. The same language is also a witness that UHAT over sequences of numbers (i.e. our setting) cannot even recognize some $AC^0$ language.
>
> While it is true that likely not every $TC^0$ language is recognized by a UHAT, we do show in the proof of Prop. 10 that UHAT recognize a $TC^0$-complete language. Thus, the $TC^0$ upper bound is best-possible in terms of complexity classes.
>
> We will clarify this in the paper.
>
> **Q: Would the paper also fit at a computer science theory conference?**
>
> A: We agree that the paper contains many theoretical results and thus would also be a candidate for a general theory venue. However, virtually all prior work on Formal Languages and Neural Networks (FLaNN) has appeared in ML and NLP venues (ICML, ICLR, ACL, EMNLP, TMLR). In addition, we believe that our results specifically clarify aspects of transformers that are of significant practical interests (see our comments on practical implications in the global rebuttal).
>
> **Q: Which logic precisely captures UHAT?**
>
> A: This is actually still an open question even for the case of finite alphabets. Barcelo et al. [3] showed that first-order logic (equivalently LTL) with monadic numerical predicates (called LTL(Mon)) is subsumed in UHAT with arbitrary position encodings, i.e., the transformer model that we are generalizing in this paper to sequences of numbers. This does not capture the full UHAT, e.g., palindrome. As remarked in [3], the logic can be extended with arbitrary linear orders on the positions (parameterized by lengths), which can capture palindrome. The same can be done for $(LT)^2L$. However, it is possible to show that the resulting logic is still not general enough to capture the full generality of UHAT.
>
> That said, although our logic does not capture the full UHAT, it can still be used to *see at a glance* what languages can be recognized by UHAT (see an example in response in global rebuttal).
>
> There could perhaps be a hope of obtaining a precise logical characterization if we *restrict* the model of UHAT. The recent paper [2] showed that LTL(Mon) captures precisely the languages of masked UHAT with position encodings with *finite image*. It is interesting to study similar restrictions for UHAT in the case of sequences of numbers.
>
> We will add the above remarks in the paper.
>
> **Q: What are the practical implications?**
>
> A: See answer in global rebuttal.

---

> > ### Comment · Reviewer_TvTa · 2024-08-08
> > **Rebuttal Read**
> >
> > I have read the rebuttal. Thank you for your responses. There are indeed interesting open questions here related to the logic. The "application" related comment serves as a good additional motivation and should preferably be added to the paper.

---

### Official Review · Reviewer_exed · 2024-07-12

**Soundness:** 4
**Presentation:** 3
**Contribution:** 3
**Rating:** 6
**Confidence:** 3

**Summary:**

The paper studies the computational expressiveness of unique hard attention transformers (UHAT) on formal languages formed over an infinite alphabet. The work is motivated by the application of transformers to time series forecasting where input values can be unbounded. Specifically, the authors assume the language to be formed over tuples of rational numbers. To this end, the paper contributes three novel results: (1) The languages recognized by UHAT belong to the circuit complexity class $\mathsf{TC}^0$ and there exists a language recognized by UHAT which is $\mathsf{TC}^0$-hard. (2) There exists a non-regular language over the alphabet $\mathbb{Q}^d$ that is recognized by a UHAT with past masking and no positional encoding. (3) UHAT with positional encoding recognize all languages expressible in an extension linear-time temporal logic with unary numerical predicates and linear rational arithmetic.

**Strengths:**

- To the best of my knowledge, the paper is the first to connect the formal language theory over infinite alphabets with the computation expressiveness of transformers. For unique hard attention transformers, a comprehensive set of results is established.
- The paper's results are particularly interesting in the light of existing results for finite alphabets. For finite alphabets, the languages accepted by UHAT are contained in $\mathsf{AC}^0$ and are exactly the star-free regular languages when masking is applied. Therefore the results provide insights into the increase in computational expressiveness due to an infinite alphabet.
- The paper’s technical claims are well presented. For each claim, either a full proof or a proof sketch with references to a full proof in the appendix is provided.
- Although the paper is densely written the contributions are clear and the paper is easy to navigate.

**Weaknesses:**

- There seems to be an important restriction to the first result: because real precision is assumed for UHAT, language inclusion can only be shown for words up to a particular length n. I assume that without this restriction the two classes are actually incomparable. At the same time, the authors seem to suggest in the concluding remarks that some results still hold when assuming rational precision for UHAT. The importance and implications of this choice should be discussed more clearly in the paper.
- The paper does not clearly convey the practical implications of the results. The infinite precision of rational numbers can not be represented in real-world systems. Instead, they would be approximated by floating point numbers such that the result of previous work applies.
- Minor: It would be helpful if the abstract already states that arbitrary input refers to the rational numbers, to avoid confusion with a real-valued input.

**Questions:**

Do the same results hold for formal languages over the alphabet $\mathbb{Q}$? If yes, why are the results derived in terms of tuples?

**Limitations:**

The authors have addressed limitations adequately, except for the real precision of UHAT. As detailed in the review it seems that some results only hold for inputs up to a particular length n. The theorems themselves do not state this.

---

> ### Author Rebuttal · Authors · 2024-08-07
>
> **Q: Is there a restriction to input lengths up to $n$?**
>
> A: We do not restrict the language to sequences of length up to $n$.  Instead, the definition of the circuit complexity class $TC^0$ is that there is a family of circuits: namely one circuit for each input length $n$ (with Boolean and majority gates, arbitrary fan-in, bounded depth, and size polynomial in $n$) that recognizes exactly the input strings of that length $n$.
>
> Thus, providing a construction for each input length $n$ is exactly what is required to prove that the entire UHAT language belongs to the class $TC^0$. In particular, the results are shown exactly as stated.
>
> **Q: Which results continue to hold in the setting with rational parameters in the UHAT?**
>
> A: What we mention in the conclusion is that many of our results not only clarify the
> setting of real precision in the transformer parameters, but also the case of rational-precision parameters: The $TC^0$ upper bound continues to hold (because rational precision
> is a restriction). Moreover the $TC^0$ lower bound also still holds, because our
> lower bound proof (Proposition 10) yields a UHAT with rational parameters.
> Likewise, the example of non-regularity is already available for rational
> precision.
>
> **Q: Do the same results hold for formal languages over the alphabet $\mathbb{Q}$? If yes, why are the results derived in terms of tuples?**
>
> A: Yes, all our main results (Theorems 1,2,3) also hold for languages over the alphabet $\mathbb{Q}$ (in other words, when $d=1$). Let us elaborate on this. This is trivial for the $TC^0$ upper bound in Theorem 1 and for Theorem 3. Moreover, our lower bound results (the non-regular example in Theorem 2, and the non-containment in $AC^0$ in Theorem 1) hold in the case of $d=1$: For the $TC^0$ lower bound (and non-containment in $AC^0$) for UHAT with positional encoding, one can use our result in Section 5 to see that the language of all length-2 sequences $(r,s)$ over $\mathbb{Q}$ with $r>s$ is accepted by a UHAT with positional encoding.
> The reason we consider the more general case of tuples is two-fold: (1) this is standard in the literature on formal languages expressiveness of UHAT: Each token is encoded by a vector of real numbers, (2) time series in general is a sequence of tuples of numbers (e.g. for stock application, a position in the sequence could be associated with max/min and entry/closing prices for the day, as well as other information like trading volume on that day).
>
> **Q: What are the practical implications?**
>
> A: See answer in global rebuttal.
>
> **Q: The infinite precision of rational numbers can not be represented in real-world systems. Instead, they would be approximated by floating point numbers such that the result of previous work applies.**
>
> A: Firstly, our result implies also that UHAT with only floating point numbers in the input is also contained in $TC^0$ and therefore is efficiently parallelizable. Secondly, although we end up with a finite alphabet if we assume a finite set of floating point numbers, this finite alphabet is extremely large ($2^{64}$ or sometimes $2^{128}$ or more in modern computers). Treating them as finite alphabets and using constructions for UHAT over finite alphabets yield extremely large $TC^0$ (in fact $AC^0$) circuits, i.e., of size at least $2^{64}$ or $2^{128}$. This is because the finite-alphabet constructions assume the alphabet size to be *constant*, meaning the actual size does not impact that complexity analysis. This is analogous to representing boolean circuits/formulas by their lookup tables, or solving games like chess/go by precomputing lookup tables (because there are finitely many configurations), none of which are realistic settings.

---

> > ### Comment · Reviewer_exed · 2024-08-12
> >
> > I thank the authors for their detailed rebuttal, in particular for clarifying on the input length $n$. This was indeed a misunderstanding on my part.

---

### Official Review · Reviewer_iac6 · 2024-07-13

**Soundness:** 4
**Presentation:** 3
**Contribution:** 3
**Rating:** 6
**Confidence:** 4

**Summary:**

This paper studies the expressive power of transformer encoders with leftmost-hard attention and strict past-masking, where the input is not a sequence of symbols from a finite alphabet, but a sequence of rational vectors. The three main results are:

1. Transformers (under the assumptions above, with or without position encodings) are in TC0 but not in AC0.
2. There is a UHAT that recognizes a non-regular language (that is, cannot be accepted by a finite automaton whose transitions are labeled with arithmetic constraints instead of symbols).
3. Every language definable in locally testable linear temporal logic (that is, linear temporal logic where an atomic formula can check a linear constraint on the current “symbol” and k lookahead “symbols”) is recognizable by a UHAT with position encoding.

**Strengths:**

Theorem 1: This proof (because the transformer weights are real) looks ambitious and interesting, although I was not able to check the details.

Theorem 2: This proof looks correct.

Theorem 3: Although I didn’t check every line of this proof, I definitely agree that the result is correct.

**Weaknesses:**

Theorem 1: I didn’t exactly understand why the inputs are rational vectors but the parameters are allowed to be real. Appendix A shows why allowing reals makes a difference, but:
(a) It doesn’t explain why the difference is important.
(b) It doesn’t show that language (2) requires real parameters, only that a particular UHAT requires real parameters to recognize language (2).
(c) One could make exactly the same argument about UHATs for strings over a finite alphabet.

With rational weights, I take it Proposition 9 would be fairly easy.

How do you define Boolean circuits that take sequences of rational vectors as input? This is hinted at in line 273 but should be spelled out more explicitly for Proposition 10 and its proof to be clear.

Theorem 2: I think it would be helpful not to reuse variable names.

Theorem 3: Just a minor comment on the phrase “logical language” in the title of Section 5. I understand a “logical language” to be the syntax of a logic, not the set of strings models of a logical sentence, so I find this phrase confusing.

**Questions:**

Could you explain why it's important for the parameters to be real, not just rational?

How do you define Boolean circuits that take sequences of rational vectors as input?

Could you expand on the significance, in this setting, of the classes AC0, regular, and TC0? Perhaps it would be helpful to give some examples of languages that do/don’t belong to these classes and expand on the practical implications. You allude to the language at lines 321-324 being important, but don’t explain why.

**Limitations:**

Yes

---

> ### Author Rebuttal · Authors · 2024-08-07
>
> **Q: What is the role of past masking in the paper?**
>
> A: Our paper mostly does *not* use masking, but instead permits arbitrary position encodings. This follows the classic model of UHAT formalized by Hao et al. [18], and has been used in various papers (e.g. see [3]). We used masking to obtain a stronger bound in the paper (e.g. Theorem 2).
>
> More precisely, masking has been used in several papers because it is a very mild version of position encodings (e.g. in the recent paper [2]). In particular, masking (future/past) can be easily simulated by using arbitrary position encodings. Without masking and position encodings, transformers cannot tell the ordering of the elements in the input sequence [27].
>
> We show in Theorem 2 that UHAT with "very mild position encodings" (namely, past masking) can already recognize non-regular languages. This is in stark contrast to the results over finite alphabets, where UHAT with masking recognize only regular languages.
>
> **Q: Why is it important to have rational inputs, but real parameters?**
>
> A: The use of rational numbers in the input sequence is standard when studying symbolic computation involving reals because (1) they can represent, among others, real numbers with finite precision, and (2) we cannot represent all real numbers by finite means.
>
> We use real parameters in the specification of transformers (e.g. in the position encodings, in the specification of affine transformations, etc.) for two reasons. Firstly, the purpose of this paper is *not* to create a new theory from scratch, but rather to extend existing works. In particular, the classic model of UHAT over finite alphabets [18] permits arbitrary real-valued parameters for affine transformations, position encodings, etc. See also the recent survey by Strobl et al. [33]. Such usage of real numbers in the formal model of transformers has been argued by the need of practical applications of transformers that also employ real-valued functions in position encodings like sin/cosine.
>
> Secondly, many of our results (e.g. that UHAT is in $TC^0$ and there is a UHAT for a $TC^0$-hard language) apply also to the more limited setting with only rational parameters.
>
> **Q: Why does Appendix A not show that any UHAT for language (2) requires real parameters?**
>
> A: The point of Appendix A is *not* to show that real parameters yield more expressiveness than rational parameters (for this one can argue in a different way, see the next question).
>
> The point of Appendix A is to illustrate the difficulty of obtaining Theorem 1: It demonstrates that proving a $TC^0$ upper bound cannot be done by showing that each real parameter in a UHAT can be replaced by a rational one while preserving recognized input sequences (even those of length 3).
>
> We overcome this difficulty by showing that one can translate the UHAT into a carefully chosen data structure (polynomial constraints with alternation bounds).  Here, the key observation is that in polynomial constraints, one *can* sufficiently approximate all real parameters by rational numbers.
>
> **Q: Why are UHAT with real parameters more expressive than UHAT with rational parameters?**
>
> A: Here is a simple example: For every real number r, one can easily build a UHAT that recognizes all sequences of length 1 and with d=1 (i.e. every accepted sequence consists of a single rational number) such that a number x is accepted if and only if x>r.  This yields a different language for each of the uncountably many choices of r.  However, there are only countably many languages with sequence length 1 and d=1 recognized by rational-parameter UHAT (since the set of rational numbers is countable).  Thus, there must be a number r for which our real-parameter UHAT has no rational-parameter equivalent.
>
> We are happy to add this to the paper.
>
> **Q: One could make exactly the same argument of Appendix A for strings over a finite alphabet.**
>
> A: Yes, exactly: The example in Appendix A shows that also for finite alphabets, one cannot sufficiently approximate the real parameters in the UHAT by rational ones.
>
> However, in the finite-alphabet case, this does not add to the difficulty, because there is no need to replace real numbers by rationals at all:  Since the alphabet is finite, there is only a finite set of intermediate values. This means, all computations can be made symbolically using a fixed table, because one only needs to distinguish the finitely many possible values.
>
> In other words, yes, Appendix A shows in particular that a naive approach to achieving sufficient precision (which is needed in the infinite-alphabet case) would already fail in the finite-alphabet case (but it is not needed there).
>
>
> **Q: How do you define Boolean circuits with rational vectors as input?**
>
> A: The rational vector is encoded as a string, where each rational number is
> encoded as a pair of binary-encoded integers. We will make this more explicit.
>
>  **Q: Significance of $AC^0$, regular and $TC^0$ in the context of UHAT over sequences. Practical implications?**
>
> A: See answer in global rebuttal.

---

> > ### Comment · Reviewer_iac6 · 2024-08-10
> >
> > Thanks for your responses!

---

### Author Rebuttal · Authors · 2024-08-07

We thank the reviewers for their in-depth reviews and useful feedback. There are some common questions with regards to practical implications of the results, which we will address here. Other questions are addressed directly to the reviewers.

**Q: What are the practical implications?**

A: We believe that the results in the paper have several interesting practical implications and applications. We mention them below, and will add them in the paper.

Firstly, that UHAT over sequences of numbers are still contained in $TC^0$ provides a justification that over sequences of numbers (be they represented as rationals or floats) are still efficiently parallelizable (more precisely, constant-time parallel complexity). This is in stark contrast to Recurrent Neural Networks, which are in the class $NC^1$ and so have logarithmic parallel complexity, and so are not as efficiently parallelizable. Note that $TC^0$ is contained in $NC^1$ (containment is not known to be strict) and that RNN can recognize $NC^1$-complete languages.

Secondly, our $TC^0$ bound can be used to understand possible limitations of UHAT in expressing various concepts over sequences of numbers. In particular, some numerical analysis concepts (e.g. SQRTSUM – which we mentioned in the paper on line 54 – determinants, and permanents) might be difficult to capture.

Thirdly, our logic $(LT)^2L$ provides a declarative language for a sufficiently large subset of UHAT. It can be used to show that some important concepts in time series can precisely be captured using UHAT. For example, take the concept of 7-day Simple Moving Average (7-SMA); this can be generalized to larger sliding windows of 50-days, or 100 days, which are often used in finance. Using $(LT)^2L$, it is easy to show that the following notion of "uptrend" can be captured using UHAT: sequences of numbers such that the value at each time t is greater than the 7-SMA value at time t. The $(LT)^2L$ formula for this is

$G( X^7\top \to \varphi(x_1,\ldots,x_7))$

where $\varphi(\bar x)$ is the formula $7x_7 > \sum_{i=1}^7 x_i$.

**Q: Significance of AC0, regular and TC0 in the context of UHAT over sequences.**

A:  That UHAT over sequences of numbers can recognize non-$AC^0$ and non-regular languages, and that UHAT are in $TC^0$ entail that the model is sufficiently powerful in performing arithmetics over supplied numbers in the input sequence, which is crucial for many applications involving sequences of numbers (e.g. time series). The latter also entails efficient parallelizability. In particular, $AC^0$ is known to have limited counting and arithmetic ability, e.g., multiplications and PARITY (the number of occurrences of a certain element in the sequence is even). *Regular languages* over sequences of numbers (i.e. well-known notion captured by symbolic automata discussed in the CACM paper [10]) have limited abilities in comparing numbers at *different positions* in the input sequence. That UHAT contains non-regular languages entails that UHAT can compare numbers at different positions in the input sequence. Finally, the circuit complexity class $TC^0$ equips $AC^0$ with the power of performing arithmetics. That UHAT over sequences of numbers capture $TC^0$-hard problems like multiplications shows the expressive power of UHAT in performing arithmetics over the numbers in the input sequence. In addition, $TC^0$ is associated with the class of languages that are efficiently parallelizable (i.e. constant time parallel complexity), as we already remarked above.

---

### Decision · Program_Chairs · 2024-09-25

**Decision:**

Accept (poster)

**Comment:**

All reviewers agree that the submission makes a significant contribution on revealing the expressive power of transformers through the lens of formal languages, and that several important findings are drawn. Although all reviewers pointed out some potential weaknesses, all of them are addressed by the authors in rebuttal. All reviewers seem satisfactory on the responses and give positive confirmations, leaving only some comments on improving the presentation which are easy to handle. Considering the significant theoretical contributions of the submission, a spotlight acceptance is recommended.